# Increased longevity due to sexual activity in mole-rats is associated with transcriptional changes in the HPA stress axis

Arne Sahm[1]*, Matthias Platzer[1], Philipp Koch[2], Yoshiyuki Henning[3], Martin Bens[4], Marco Groth[4], Hynek Burda[5,6], Sabine Begall[5], Saskia Ting[7], Moritz Goetz[7], Paul Van Daele[8], Magdalena Staniszewska[9], Jasmin Mona Klose[9], Pedro Fragoso Costa[9], Steve Hoffmann[1†], Karol Szafranski[2†], Philip Dammann[5,10†]

[1]Computational Biology Group, Leibniz Institute on Aging – Fritz Lipmann Institute, Jena, Germany; [2]Core Facility Life Science Computing, Leibniz Institute on Aging – Fritz Lipmann Institute, Jena, Germany; [3]Institute of Physiology, University Hospital, University of Duisburg-Essen, Essen, Germany; [4]Core Facility Sequencing, Leibniz Institute on Aging – Fritz Lipmann Institute, Jena, Germany; [5]Department of General Zoology, Faculty of Biology, University of Duisburg-Essen, Essen, Germany; [6]Department of Game Management and Wildlife Biology, Faculty of Forestry and Wood Sciences, Czech University of Life Sciences, Prague, Czech Republic; [7]Institute of Pathology and Neuropathology, University Hospital, University of Duisburg-Essen, Essen, Germany; [8]Department of Zoology, University of South Bohemia, České Budějovice, Czech Republic; [9]Department of Nuclear Medicine, University Hospital, University of Duisburg-Essen, Essen, Germany; [10]Central Animal Laboratory, University Hospital, University of Duisburg-Essen, Essen, Germany

*For correspondence:
arne.sahm@leibniz-fli.de

†These authors contributed equally to this work

Competing interests: The authors declare that no competing interests exist.

**Abstract** Sexual activity and/or reproduction are associated with a doubling of life expectancy in the long-lived rodent genus *Fukomys*. To investigate the molecular mechanisms underlying this phenomenon, we analyzed 636 RNA-seq samples across 15 tissues. This analysis suggests that changes in the regulation of the hypothalamic–pituitary–adrenal stress axis play a key role regarding the extended life expectancy of reproductive vs. non-reproductive mole-rats. This is substantiated by a corpus of independent evidence. In accordance with previous studies, the up-regulation of the proteasome and so-called 'anti-aging molecules', for example, dehydroepiandrosterone, is linked with enhanced lifespan. On the other hand, several of our results are not consistent with knowledge about aging of short-lived model organisms. For example, we found the up-regulation of the insulin-like growth factor 1/growth hormone axis and several other anabolic processes to be compatible with a considerable lifespan prolongation. These contradictions question the extent to which findings from short-lived species can be transferred to longer-lived ones.

## Introduction

Most of our current understanding of the underlying mechanisms of aging comes from short-lived model species. It is, however, still largely unclear to what extent insights obtained from short-lived organisms can be transferred to long-lived species, such as humans (*Parker et al., 2004*; *Keller and Jemieliy, 2006*). Comparative approaches, involving species with particularly long healthy lives and

seeking the causative mechanisms that distinguish them from shorter-lived relatives, try to overcome this limitation (*Austad, 2009*). Many studies that involved organisms with particularly long lifespans, for example, queens in social hymenoptera, birds, bats, African mole-rats, and primates, have produced findings that were not always congruent with established aging theories (*Keller and Jemielity, 2006*; *Austad, 2009*; *Salmon et al., 2009*; *Austad, 2011*; *Dammann, 2017*; *Bens et al., 2018*). Species comparisons, however, also have their limitations. Many observed differences between species with differing lifespans are influenced by phylogenetic constraints, ecophysiological differences, or both, rather than being causal for the species-specific differences in aging and longevity.

Bimodal aging occurs naturally in the genus *Fukomys* from the rodent family Bathyergidae (African mole-rats). These animals live in families (often called colonies) of usually consisting of 9–16 individuals (*Sichilima et al., 2008*; *Šklíba et al., 2012*), although single families may occasionally grow considerably larger in some species (*Jarvis and Bennett, 1993*; *Scharff et al., 2001*). Regardless of group size, an established family typically consists of only one breeding pair (the founders of the family, often called king/queen) and their progeny from multiple litters (often called workers). Because of strict avoidance of incest (*Burda, 1995*), the progeny do not engage in sexual activity in the confines of their natal family, even after reaching sexual maturity. Hence, grown *Fukomys* families are characterized by a subdivision into breeders (the founder pair) and non-breeders (all other family members). Interestingly, breeders reach the age of 20 years or more in captivity, whereas non-breeders usually die before their tenth birth date (*Figure 1A*). This divergence of survival probabilities between breeders and non-breeders is found in all *Fukomys* species studied so far, irrespective of sex. Because no difference in diet or workload has been observed between breeders and non-breeders in captivity, status-specific changes of gene expression after the transition from non-breeder to breeder are considered the most likely explanation of the differing lifespans (*Dammann and Burda, 2006*; *Dammann et al., 2011*).

In the wild, non-breeders must meet a member of another family by chance to ascend to breeder status; in captivity, the establishment of new breeder pairs is subject to human control. Allowing an animal to breed in captivity can be regarded as a simple experimental intervention that results in an extension of life expectancy of approximately 100%. This extension is far more than most experimental interventions in vertebrates can achieve, for example, by caloric restriction (e.g., *Carmona and Michan, 2016*) or diets containing resveratrol or rapamycin (e.g., *Valenzano et al., 2006*; *Johnson et al., 2013*). Furthermore, this relative lifespan extension starts from a non-breeder lifespan that is already more than twice as long as that of the mammalian model organisms most widely used in aging research, such as mice or rats.

Until now, relatively few studies have addressed the potential mechanisms behind this natural status dependency of aging in *Fukomys* sp. Contrary to the predictions of both the advanced glycation formation theory (*Morimoto and Cuervo, 2009*) and the oxidative stress theory of aging (*Harman, 1956*; *Harman, 2001*), markers of protein cross-linking and -oxidation were surprisingly higher in breeders of Ansell's mole-rats (*Fukomys anselli*) than in age-matched non-breeders (*Dammann et al., 2012*). On the other hand, in the Damaraland mole-rat (*Fukomys damarensis*), oxidative damage to proteins and lipids was significantly lower in breeding females than in their non-reproductive counterparts (*Schmidt et al., 2014*), a finding that is compatible with the oxidative stress theory of aging. In good agreement with their overall longevity irrespective of social status, Ansell's mole-rats produce less thyroxine (T4) and recruit smaller proportions of their total T4 resources into the active unbound form than do euthyroid mammals. Still, nonetheless the levels of unbound T4 (fT4) do not explain the intraspecific longevity differences between *F. anselli* breeders and non-breeders because the levels of this hormone did not differ between the two cohorts (*Henning et al., 2014*). Closely connected to the topic of this paper is the finding that non-breeding giant mole-rats (*Fukomys mechowii*) maintain fairly stable gene expression into relative old ages, quite in contrast to the shorter-lived Norway rat (*Sahm et al., 2018a*). It is, however, still unclear what happens on the gene expression level when an individual attains breeding status.

Interestingly, a recent study by *Bens et al., 2018* found that the longest-lived rodent, the naked mole-rat *Heterocephalus glaber*, tended globally to show opposite changes in the transition from non-breeders to breeders compared to shorter-lived guinea pigs. *Heterocephalus* and *Fukomys* are similar in their mating and social behavior, but differences appear to exist regarding the effect of breeding on longevity: until very recently, naked mole-rat non-breeders have been reported to be

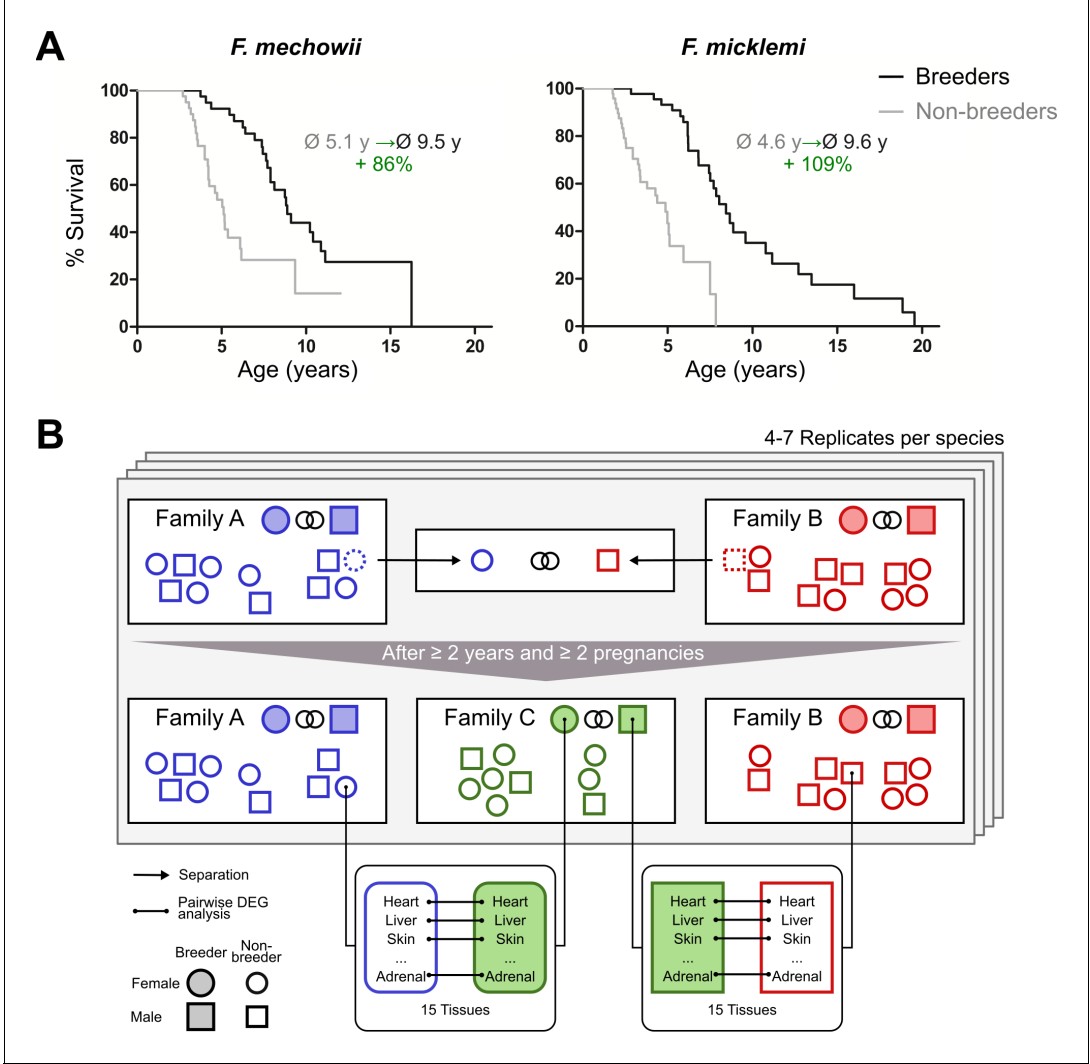

**Figure 1.** Motivation (A) and principle of the experimental setup (B). (A) For both species of the *Fukomys* genus that were examined in this study – *Fukomys mechowii* and *Fukomys micklemi* – it was shown that, in captivity, breeders live significantly longer than non-breeders. Lifespan data were redrawn from *Dammann and Burda, 2006* and *Dammann et al., 2011*. (B) Schematic overview of animal treatments. Non-breeders (open shapes) are offspring of the breeder pair of their family (filled shapes) and do not mate with other members of the same family because of incest avoidance in *Fukomys* (*Burda, 1995*). Therefore, non-breeders of opposite sexes were taken from different families – labeled as 'Family A' (blue) and 'Family B' (red) – and permanently housed in a separate terrarium. The two unrelated animals mated with each other, thus producing offspring and becoming breeders of the new 'Family C' (green). In addition to the animals that were promoted to be slower-aging breeders, age-matched controls that remained in the faster-aging non-breeders of 'Family A' and 'Family B' were included in our study – in most cases full siblings (ideally litter mates) of the respective new breeders. After at least 2 years and two pregnancies in 'Family C', breeders from 'Family C' and their controls from Colonies A and B were put to death, and tissues were sampled for later analysis, which included identification of differentially expressed genes. The shown experimental scheme was conducted with 5–7 (median 6) specimens per cohort (defined by breeding status, sex, species) and 12–15 tissues (median 14) per specimen: in total, 46 animals and 636 samples.

The online version of this article includes the following figure supplement(s) for figure 1:

**Figure supplement 1.** Overview of the tissues examined in this study with a schematic mole-rat representation.

as long-lived as breeders (*Sherman and Jarvis, 2002*; *Buffenstein, 2008*). In 2018, a lifespan advantage of breeders over non-breeders was reported in females (but not males), yet the divergence of the two groups appeared to be considerably smaller than in *Fukomys* (*Ruby et al., 2018*) and underlying data is being debated (*Dammann et al., 2019*). In summary, status-dependent aging is either absent in *Heterocephalus* or less pronounced than in *Fukomys*. Nevertheless, both *Fukomys* and naked mole-rats can be seen as counterexamples to the disposable soma theory of aging, which

assumes that the aging process is linked to an evolutionary tradeoff in which the preservation of the organism is essentially sacrificed to reproductive success (*Kirkwood, 1977*).

To gain insights into the molecular basis of the bimodal aging pattern in *Fukomys*, it would be of course interesting to compare sufficiently large cohorts of old breeders and non-breeders both at similar chronological and biological ages. But as old animals, particularly of a long-lived species, are extremely valuable due to obvious logistical, timely, and financial reasons, such an approach was impossible for us. As an alternative, we decided to study young breeders and non-breeders that are much easier to obtain and hypothesized that the status change marks the beginning of a slowdown in the aging process. In this paper, we studied two *Fukomys* species (*F. mechowii* and *Fukomys micklemi*, *Figure 1A*) by comparing the gene expression profiles of breeders (n = 24) and age-matched non-breeders (n = 22) in 16 organs or their substructures (hereinafter referred to as tissues, *Figure 1—figure supplement 1*, *Figure 1B*). Our main aim was to identify genes and pathways whose transcript levels are linked to the status-dependent longevities and to relate these patterns to insights into aging research obtained in shorter-lived species.

## Results

We measured gene expression differences between breeders and non-breeders in two African mole-rat species, *F. mechowii* and *F. micklemi*. Altogether, we performed RNA-seq for 636 tissue samples covering 16 tissue types from both species, sexes, and reproductive states (breeders and non-breeders). Each of the four groups (male/female breeders/non-breeders) of each species consisted of 5–7 animals (see *Supplementary file 1a–e* for sample sizes, animal data, and pairing schemes). For each tissue, we conducted a multifactorial analysis of differentially expressed genes (DEGs): the analysis was based on the variables reproductive state, sex, and species. During this exercise, we focused on the differences between slower-aging breeders and faster-aging non-breeders. This approach increases our statistical power by giving us a fourfold increase of sample size in comparison to species- and sex-specific breeder vs. non-breeder analyses. At the same time, we can additionally reduce the number of false-positive DEGs by restricting the analysis to those breeding status-related genes that show the same direction in both sexes and species. We deliberately focused on those genes to concentrate our study on universal mechanisms that hold for both sexes and species. The number of samples collected and processed for this work required splitting them into different batches for sequencing. To analyze the degree this procedure potentially biased our results, we systematically investigated potential batch effects. Our results suggest that the sequencing strategy has little effect on the outcomes reported in the following (see Control analyses).

To globally quantify the transcriptomic differences between the reproductive states, we performed three analyses: clustering of the samples based on pairwise correlation, principal variant component analysis, and an overview of the number of DEGs between reproductive states in comparison to DEGs between species and sex. Clustering of the samples based on pairwise correlations showed a full separation of the two species at the highest cluster level (*Figure 2—figure supplement 1*). Below that level, an almost complete separation according to tissues was observed. Within the tissue clusters, the samples did not show a clear-cut separation between sex or breeder/non-breeder status. Accordingly, a principal variance component analysis showed that species, tissue, and the combination of both variables accounted for 98.4% of the total variance in the data set; individual differences explained 1.4% of the variance, and only 0.004% was explained by breeder/non-breeder status (*Figure 2A*). Regarding the numbers of DEGs, we found – unsurprisingly considering the aforementioned facts – by far the highest number of DEGs in the species comparison (*Figure 2B*). Although in almost every examined tissue the numbers of detected DEGs were also high between sexes, most tissues exhibited very few DEGs due to breeder/non-breeder status. Exceptions were liver, spleen, ovary, and, especially, tissues of the endocrine system (adrenal gland, pituitary gland, thyroid), in which the number of DEGs between breeders and non-breeders ranged from more than 60 to several thousand.

Since the change from non-breeder to breeder status apparently marks the beginning of a slowdown in the aging process, we first wanted to find out whether and where there are intersections of reproductive status DEGs with those whose expression level is known to change during aging. Therefore, we determined overlaps by using the Digital Aging Atlas (DAA) – a database of genes

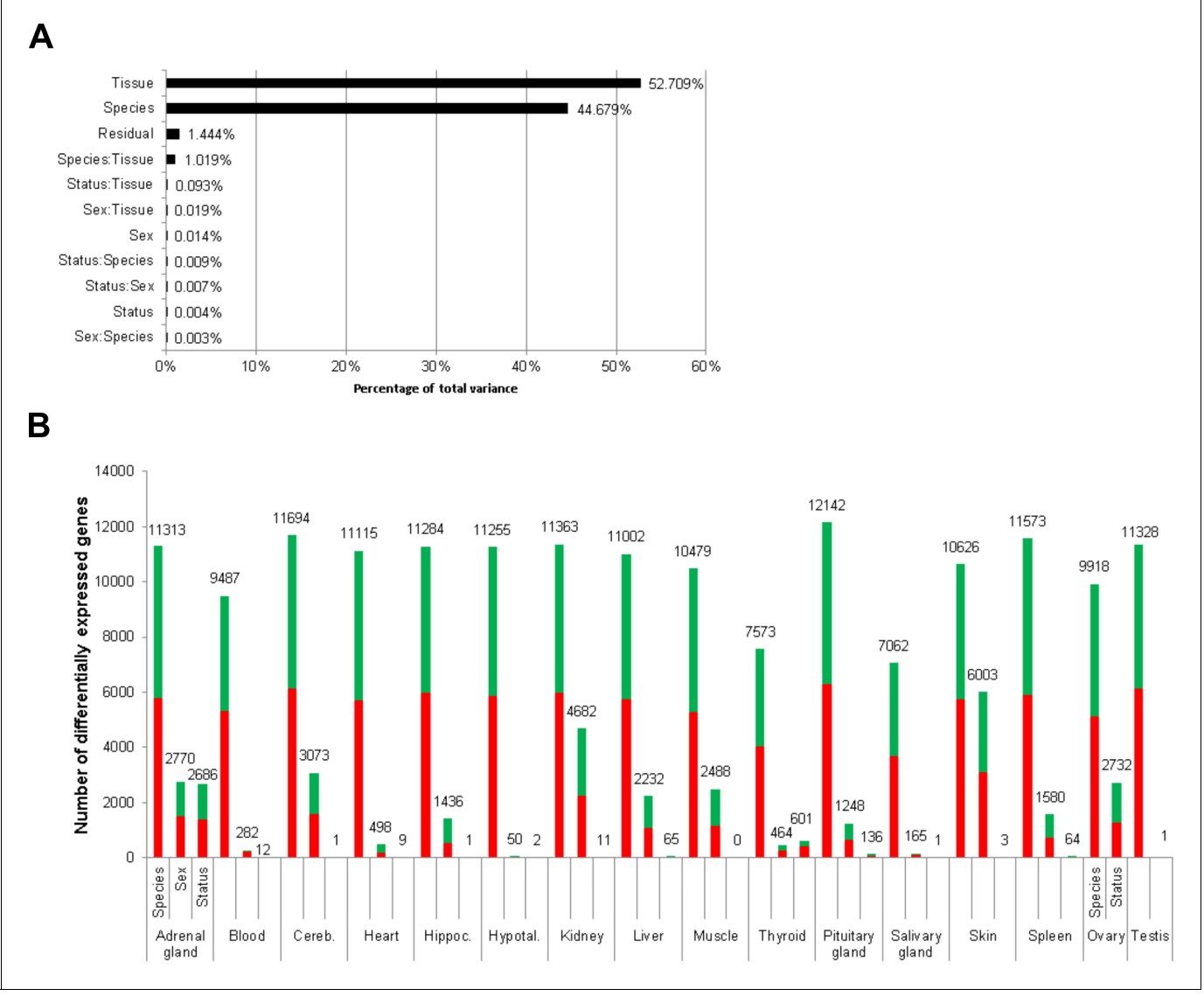

**Figure 2.** Total variance distribution (A) and numbers of differentially expressed genes (B). (A) Relative contribution of the model factors (breeding status, sex, species, tissue) and their combinations (:) to the total variance in the examined data set. The relative contributions were determined by principal variance component analysis. (B) Numbers of identified differentially expressed genes per tissue and model factor (first column, species; second, sex; third, status). Stacked bars indicate the proportions of up- and down-regulated genes (red and green, respectively; directions: *F. mechowii* vs. *F. micklemi*, female vs. male, breeder vs. non-breeder).

The online version of this article includes the following figure supplement(s) for figure 2:

**Figure supplement 1.** Clustering of the 636 examined samples.

**Figure supplement 2.** Comparison of differentially expressed genes identified by DESeq2 and limma (worker vs. breeders).

that show aging-related changes in humans (*Craig et al., 2015*). Across species and sexes, significant overlaps (false discovery rate [FDR] < 0.05, Fisher's exact test) with the DAA were found in three tissues: adrenal gland, ovary, and pituitary gland (FDR=0.005, each; *Figure 3A*). Among these three endocrine tissues, the DEGs of the ovaries overlapped significantly with those from adrenal (p=2.8*10$^{-27}$) and pituitary glands (p=0.005), but there was no significant overlap between the two glands (*Figure 3A*). Thus, together, we found indications for aging-relevant expression changes after the transition from non-breeders to breeders in three tissues of the endocrine system, which presumably affect separate aspects of aging in adrenal and pituitary glands.

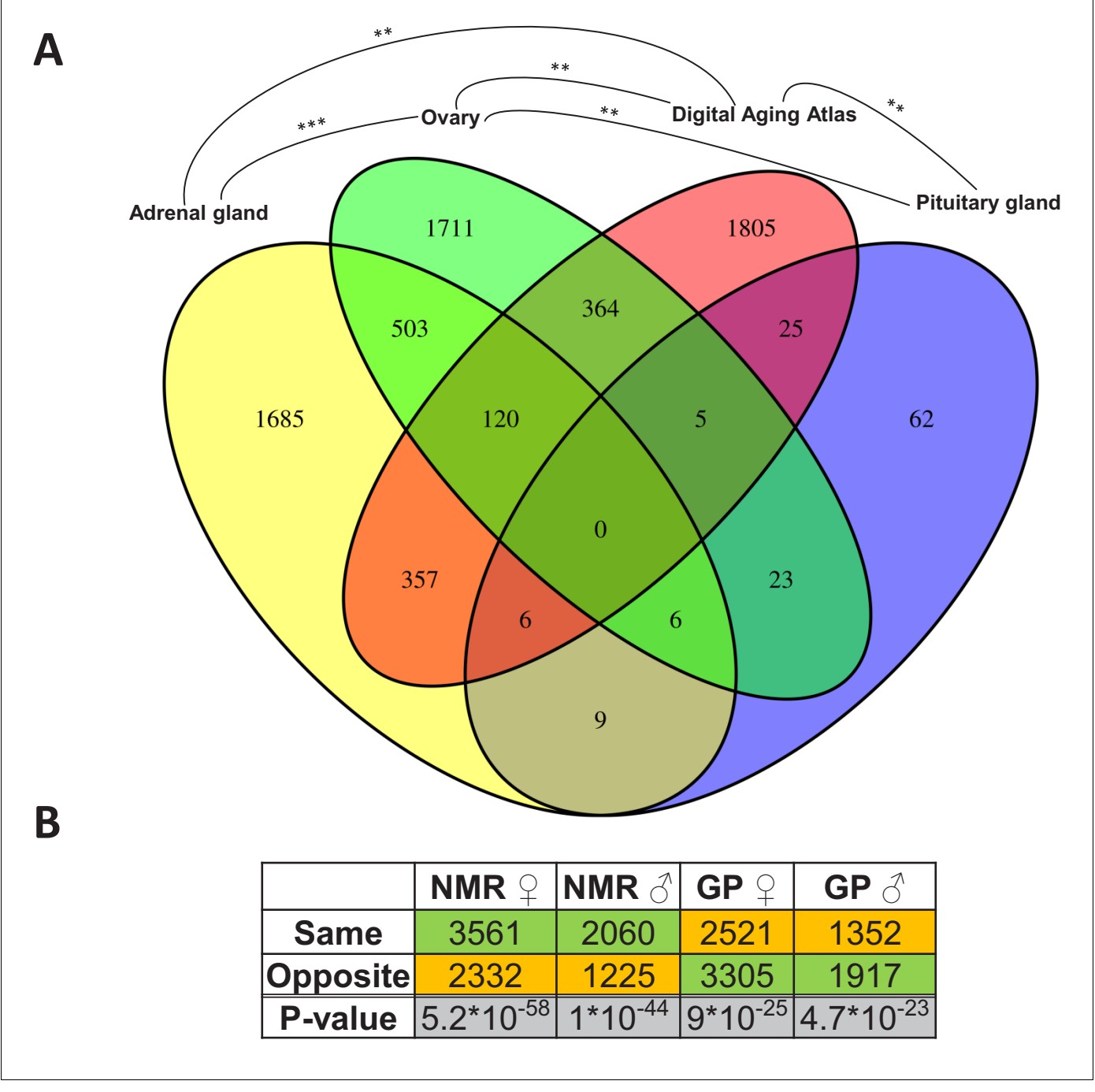

**Figure 3.** Assessment of the aging relevance of genes that are differentially expressed between breeders and non-breeders. (**A**) For each tissue, we separately tested whether the identified differentially expressed genes between status groups significantly overlapped with the genes within the Digital Aging Atlas database (Fisher's exact test, false discovery rate [FDR] < 0.05). Significant overlaps were found for three tissues: adrenal gland, ovary, and pituitary gland. The Venn diagram depicts the overlaps of these three tissues with the Digital Aging Atlas and with each other (**FDR < 0.01; ***FDR < 0.001). (**B**) A similar experiment comparing the transcriptomes of breeders vs. non-breeders was recently conducted in naked mole-rats (NMRs) and guinea pig (GPs) (*Bens et al., 2018*). For NMR, there is also evidence that breeders have a (slightly) longer lifespan than non-breeders, whereas for GP the opposite is assumed (*Bens et al., 2018*; *Ruby et al., 2018*). Across 10 tissues that were examined in both studies, the analysis determined whether status-dependent differentially expressed genes identified in the current study were regulated in the same or opposite direction in NMR and GP (*Supplementary file 1f*). The listed p-values (two-sided binomial exact test; hypothesized probability, 0.5) describe how extremely the ratio of genes expressed in the same and opposite directions deviates from a 50:50 ratio. Green and orange indicate the majority and minority of genes within a comparison, respectively. Figure created with BioRender.com.

Moreover, we compared the DEGs with respect to the reproductive status that we identified in *Fukomys* with regard to their direction to transcript-level changes observed in similar experiments using naked mole-rats and guinea pigs (*Bens et al., 2018*). The direction of the status-dependent DEGs regulation in *Fukomys*, as found in this study, was significantly more often the same rather than opposite as in the naked mole-rat (females, 60%, p=$5.2*10^{-58}$; males, 62%, p=$10^{-44}$ for females, *Figure 3B*, *Supplementary file 1f*). In the guinea pig, on the contrary, the *Fukomys* reproductive status DEGs were significantly more often regulated in the opposite direction (females, 57%, p=$9*10^{-25}$; males, 59%, p=$4.7*10^{-23}$, *Figure 3B*, *Supplementary file 1f*). Thus, at the single-gene level, the expression changes linked to reproductive status may affect lifespan differently in long-lived African mole-rats than in shorter-lived guinea pigs.

Beyond the single-gene level, we aimed to identify metabolic pathways and biological functions whose gene expression significantly depends on reproductive status. For this, we used Kyoto Encyclopedia of Genes and Genomes (KEGG) pathways (*Kanehisa et al., 2017*) and Molecular Signatures Database (MSigDB) hallmarks (*Liberzon et al., 2015*) as concise knowledge bases. As standard approach, we used an established method combining all p-values of genes in a given pathway in a threshold-free manner (*Figure 4*). The advantage of this approach is that it bundles the p-values from test results of individual gene expression differences at the level of pathways (see Materials and methods for details). In addition, we applied a second enrichment method that aggregates fold-changes instead (*Figure 4—figure supplement 11*; for a comparison of the results of both approaches, see Control analyses). Altogether, the gene expression of 55 KEGG pathways and 41 MSigDB hallmarks was significantly affected by reproductive status in at least one tissue (*Figure 4—figure supplements 1* and *2*). Because the individual interpretation of each of these pathways/hallmarks would go beyond the scope of this study, we focus here on those 14 pathways and 13 hallmarks that were significantly different between breeders and non-breeders (FDR < 0.1) in a global analysis across all tissues (*Figure 4*). Because many pathways are driven mainly by gene expression in subsets of tissues, we weighted gene-wise the differential expression signals from the various tissues by the respective expression levels in the tissues. For instance, the expression level of the growth hormone (GH) gene *GH1* is known to be almost exclusively expressed in the pituitary glands. In our data set, the *GH1* level of the pituitary gland accounted for 99.96% of the total *GH1* across all tissues. Accordingly, in pathways that contain *GH,* our weighted cross-tissue differential expression signal for this gene is almost exclusively determined by the pituitary gland. On the contrary, a differential expression signal of this gene in another tissue with a very low fraction of the gene's total expression would have almost no impact on the weighted cross-tissue level – even if that signal were very strong (see Materials and methods for details).

We found strong indications for increase in the activity of certain anabolic functions in breeders: ribosomal protein expression (hsa03010 Ribosome, hsa03008 Ribosome biogenesis in eukaryotes, *Figure 4A*) was elevated in most tissues (and accordingly also in the weighted cross-tissue analysis). In MSigDB hallmarks, the strongest enrichment signal came from MYC targets (HALLMARK_MYC_TARGETS_V1), which can largely be considered a reflection of enhanced ribosomal protein expression and the fact that MYC is a basal transcription factor up-regulating genes involved in protein translation (*Hofmann et al., 2015*; *Figure 4B*). In functional correspondence, we observed an increase in the expression of mitochondrial respiratory chain components (hsa00190 Oxidative phosphorylation, HALLMARK_OXIDATIVE_PHOSPHORYLATION, *Figure 4A, B*). We also found strong indications for increased protein degradation (hsa03050 PROTEASOME, *Figure 4A*). This weighted cross-tissue signal was, in contrast to the situation regarding ribosomes, driven mainly by two tissue types: the adrenal gland and the gonads.

To examine whether the simultaneous up-regulation of the ribosome, proteasome, and oxidative phosphorylation is a coordinated regulation, we performed a weighted gene co-expression network analysis (WGCNA) (*Langfelder and Horvath, 2008*) from our gene count data and examined the connectivity between pairs of those KEGG pathways flagged in the weighted cross-tissue analysis. We found that the expression of ribosomal genes (hsa03010) was significantly linked to those of ribosome biogenesis (hsa03008, FDR = $4.59*10^{-3}$), oxidative phosphorylation (hsa00190, FDR = $4.05*10^{-4}$), and proteasome (hsa03050, FDR = $4.59*10^{-3}$), whereas no other examined pathway pair exhibited a significant connectivity (*Figure 4—figure supplement 3*). Interestingly, ribosome, proteasome, and oxidative phosphorylation pathways also shared other characteristics of their differential expression signals: subtle fold-changes, that is, up-regulation of 3–9% on average.

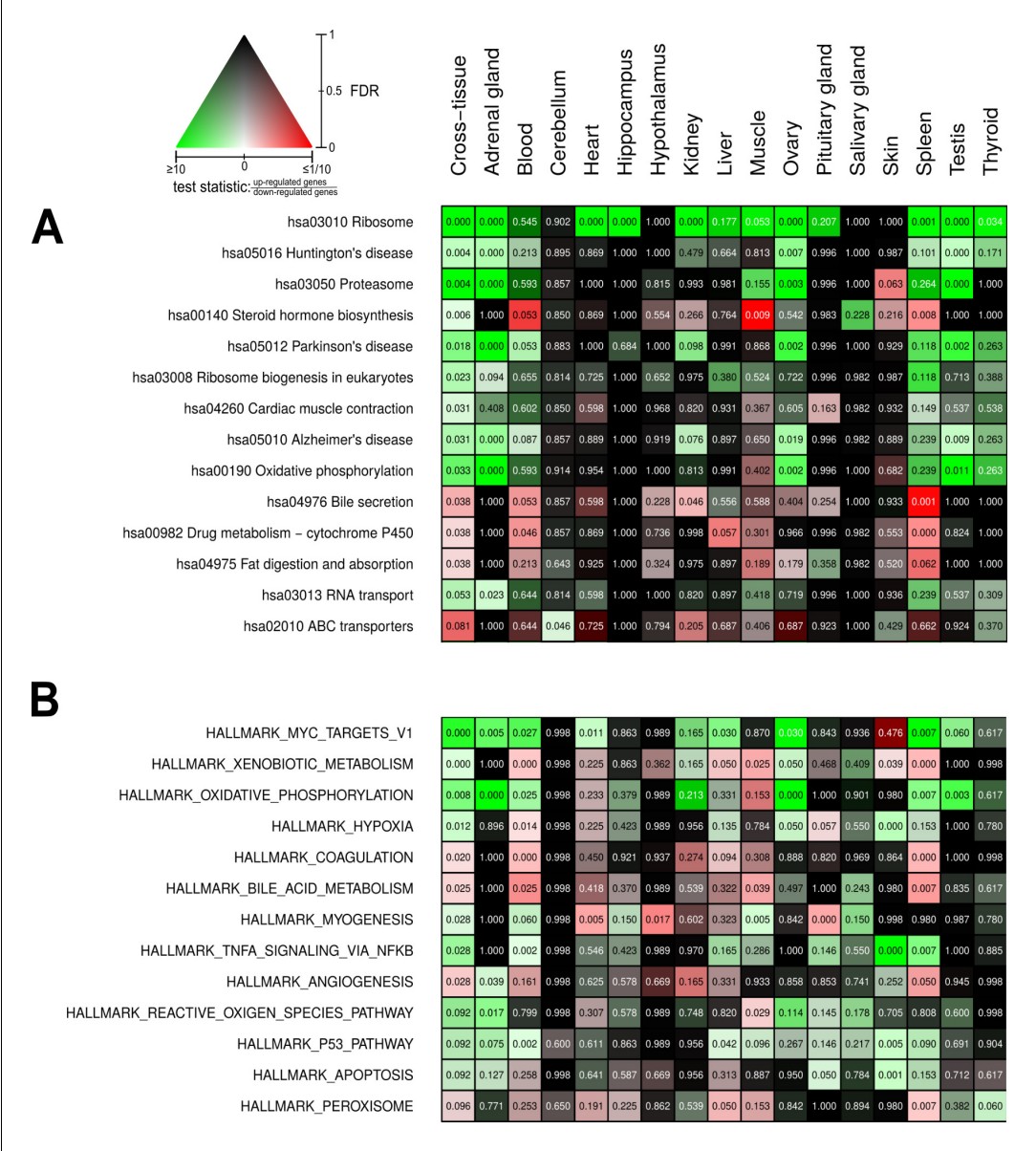

**Figure 4.** Pathways and metabolic functions enriched for status-dependent differential gene expression. Shown are all Kyoto Encyclopedia of Genes and Genomes pathways (**A**) and Molecular Signatures Database hallmarks (**B**) that are enriched for differential gene expression between breeders and non-breeders at the weighted cross-tissue level (false discovery rate [FDR], <0.1). This enrichment analysis was carried out threshold-free, which is why tissues without differentially expressed individual genes (see *Figure 2B*) can also show differentially expressed pathways. The numbers within the cells give the FDR, that is, the multiple testing corrected p-value, for the respective pathway/hallmark and tissue. As indicated by the color key, red and green indicate up- or down-regulated in breeders, respectively. White indicates a pathway/hallmark that is significantly affected by differential expression and whose signals for up- and down-regulation are approximately balanced. Dark colors up to black mean that there is little or no evidence that the corresponding pathway/hallmark is affected by differential gene expression. *Figure 4—figure supplements 1* and *2* provide detailed overviews of all pathways/hallmarks that are enriched in at least one tissue for status-dependent differential expression signals.

The online version of this article includes the following figure supplement(s) for figure 4:

**Figure supplement 1.** Full overview of Kyoto Encyclopedia of Genes and Genomes pathways that are enriched for status-dependent differential gene expression.

**Figure supplement 2.** Full overview of Molecular Signatures Database hallmarks that are enriched for status-dependent differential gene expression.

**Figure supplement 3.** Pairwise connectivity of metabolic pathways that are enriched for caste-dependent differential gene expression at the cross-tissue level.

**Figure supplement 4.** Status-dependent regulation of Kyoto Encyclopedia of Genes and Genomes pathway hsa00140 steroid hormone biosynthesis at the cross-tissue level.

*Figure 4 continued on next page*

*Figure 4 continued*

**Figure supplement 5.** Status-dependent regulation of Kyoto Encyclopedia of Genes and Genomes pathway hsa05016 Huntington's disease at the cross-tissue level.

**Figure supplement 6.** Status-dependent regulation of Kyoto Encyclopedia of Genes and Genomes pathway hsa05012 Parkinson's disease at the cross-tissue level.

**Figure supplement 7.** Status-dependent regulation of Kyoto Encyclopedia of Genes and Genomes pathway hsa050160 Alzheimer's disease at the cross-tissue level.

**Figure supplement 8.** Cross-tissue p-value vs. gene expression.

**Figure supplement 9.** Cross-tissue p-value vs. pathway expression.

**Figure supplement 10.** Cross-tissue p-value vs. pathway size.

**Figure supplement 11.** Pathways and metabolic functions enriched for status-dependent differential gene expression checked by another enrichment method.

Thus, statistically significant signals at the pathway level resulted from relatively small shifts in all genes of these pathways in a seemingly coordinated manner and across multiple tissues (*Source data 1a*). In addition, ribosome (hsa03010, in ovary), proteasome (hsa03050, in ovary and adrenal gland), and RNA-transport (hsa03013, in adrenal gland) are enriched in those *Fukomys* status-dependent DEGs that show, in a similar experimental setting (*Bens et al., 2018*), the same direction in both naked mole-rat sexes and the opposite direction in both guinea pig sexes (FDR < 0.05, Fisher's exact test).

The myogenesis hallmark (*Figure 4B*) was also found to be up-regulated in breeders. Expectedly, this weighted cross-tissue result was driven mainly by differential expression signals from muscle tissue: muscle from all tissues exhibited the lowest p-value (*Figure 4A*), and 15 of 20 up-regulated genes that contributed most to the weighted cross-tissue differential myogenesis signal exhibited their highest expression in muscle. These genes were involved mainly in calcium transport or part of the fast-skeletal muscle-troponin complex (*Source data 1a*). A clear exception of this muscle-dominated expression is found in the gene that exhibited the highest relative contribution to the differential myogenesis signal, *insulin-like growth factor 1* (*IGF1*). This gene was found to be expressed most strongly in ovary and liver and was strongly up-regulated in the breeders' ovaries and adrenal glands (*Table 1*). *IGF1* codes for a well-known key regulator of anabolic effects such as cell proliferation,

**Table 1.** Top 10 genes regarding weighted cross-tissue differential expression signal.

| | Weighted cross-tissue | | | Tissue with highest expression | | | |
|---|---|---|---|---|---|---|---|
| | p-Value | FDR | log2-foldchange* | Tissue | % of cross-tissue expression | FDR | log2-foldchange* |
| *SULT2A1* | 0.00E+00 | 0.000 | −3.19 | Liver | 97.42 | 5.27E-06 | -3.26 |
| *MC2R* | 6.00E-06 | 0.046 | −0.53 | Adrenal gland | 89.53 | 8.78E-05 | -0.56 |
| *INHA* | 1.60E-05 | 0.081 | 1.57 | Ovary | 67.91 | 1.54E-05 | 2.22 |
| INHA – tissue with second highest expression | | | | Testis | 27.61 | 0.56 | 0.21 |
| *CYP11A1* | 5.90E-05 | 0.224 | 0.61 | Ovary | 45.55 | 5.49E-05 | 0.91 |
| CYP11A1 – tissue with second highest expression | | | | Adrenal gland | 43.63 | 1.48E-03 | 0.47 |
| *NLRP14* | 1.17E-04 | 0.355 | −0.80 | Ovary | 68.98 | 2.84E-05 | −1.25 |
| NLRP14 – tissue with second highest expression | | | | Testis | 18.67 | 0.54 | 0.19 |
| *GH* | 2.58E-04 | 0.618 | 0.45 | Pituitary gland | 99.96 | 1.99E-02 | 0.45 |
| *IGF1* | 3.62E-04 | 0.618 | 0.59 | Ovary | 51.73 | 1.20E-04 | 0.76 |
| IGF1 – tissue with second highest expression | | | | Liver | 36.28 | 0.39 | 0.24 |
| *ZP4* | 3.74E-04 | 0.618 | −1.16 | Ovary | 95.30 | 1.99E-03 | −1.23 |
| *TCL1A* | 5.33E-04 | 0.618 | −1.05 | Ovary | 90.12 | 2.14E-03 | −1.18 |
| *PNLIPRP2* | 5.69E-04 | 0.618 | 1.36 | Ovary | 76.08 | 2.23E-03 | 1.65 |

*Direction: breeder/non-breeder.

Tissues are listed if they contribute at least 10% to cross-tissue expression.

FDR: false discovery rate.

myogenesis, and protein synthesis (*Schiaffino and Mammucari, 2011*; *Jung and Suh, 2014*) and has a tight functional relation to GH (gene: *GH1*) another key anabolic regulator upstream of *IGF1*; together, these factors form the so-called GH/IGF1 axis (*Cannata et al., 2010*; *Junnila et al., 2013*; *Bodart et al., 2015*; *Raisingani et al., 2017*; *Carotti et al., 2018*; *Lozier et al., 2018*). Also, *GH1* was strongly up-regulated in breeders in its known principal place of synthesis, the pituitary gland (*Table 1*).

With xenobiotic metabolism and TNF-α signaling, two defense hallmarks were also found to be up-regulated in breeders by the weighted cross-tissue analysis (HALLMARK_XENOBIOTIC_METABL-ISM and HALLMARK_TNFA_SIGNALING_VIA_NFKB, *Figure 4B*). The up-regulation of the reactive oxygen species (ROS) hallmark comprising genes coding for proteins that detoxify ROS (HALL-MARK_REACTIVE_OXYGEN_SPECIES_PATHWAY, *Figure 4B*) falls into a similar category.

Another interesting aspect that was found to be significantly altered in breeders is steroid hormone biosynthesis (hsa00140 Steroid hormone biosynthesis, *Figure 4A*). In this case, both up- and down-regulated genes were in the pathway, and their absolute fold-changes were roughly balanced. Steroid hormones, on the one hand, comprise sex steroids – because these hormones are important players in sexual reproduction, such differences should be expected given the experimental setup. On the other hand, the class of steroid hormones – corticosteroids – has regulatory functions in metabolism, growth, and the cardiovascular system, as well as in the calibration of the immune system and response to stress (*Liu et al., 2013*). The most influential contributor to the differential pathway signal by far on the weighted cross-tissue level was *CYP11A1,* which codes for the (single) enzyme that converts cholesterol to pregnenolone. This is the first and rate-limiting step in steroid hormone synthesis (*Miller and Auchus, 2011*). Because *CYP11A1* was found to be up-regulated in breeders in its main places of synthesis – the gonads and the adrenal gland (*Table 1*) – it can be assumed that the total output of steroid hormone biosynthesis in breeders is increased. The pattern of up- and down-regulation on the KEGG pathway, however, suggests that sex steroids especially are produced at a higher rate in breeders, whereas circulating levels of glucocorticoids – such as cortisol – should be lower than in non-breeders (*Figure 4—figure supplement 4*; see also our discussion on ACTH-R below).

Finally, several pathways flagged by the weighted cross-tissue analysis seem to be derivatives of the abovementioned differentially expressed pathways instead of representing altered functions on their own. For example, Huntington's (hsa05016), Parkinson's (hsa05012), and Alzheimer's (hsa05010) diseases could, in principle, be interpreted as highly relevant for aging and lifespan. A closer inspection of these pathways reveals, however, that the genes of the mitochondrial respiratory chain – which is the core of the oxidative phosphorylation pathway – are in all three cases the main contributors to the respective differential expression signals (*Figure 4—figure supplements 5–7, Source data 1a*). Similarly, we see in the case of fat digestion (hsa04975 Fat digestion and absorption) that two of the three largest contributors to the differential expression signal of that pathway – *ABCG8* and *SCARB1* – are directly involved in the transport of cholesterol (*Liu et al., 2008*; *Wang et al., 2015*). Therefore, it seems likely that this signal is an expression of the altered steroid hormone biosynthesis rather than indicating altered fat digestion.

Interestingly, three genes, which we had already mentioned as potential regulators during the pathway analysis, also appeared among the 10 most clearly altered genes on the weighted cross-tissue level: *GH1, IGF1,* and *CYP11A1* (*Table 1*). The top 2 among these 10 are *sulfotransferase family 2A member 1* (*SULT2A1*) and *melanocortin two receptor* (*MC2R*). SULT2A1 is the main catalyzer of the sulfonation of the steroid hormone dehydroepiandrosterone (DHEA) to its non-active form DHEA-S (*Hammer et al., 2005*). DHEA has repeatedly been proposed to be an 'anti-aging hormone' because its levels are negatively associated with chronological aging (*Baulieu, 1996*; *Celec and Stárka, 2003*; *Rutkowski et al., 2014*). We found that *SULT2A1* is strongly down-regulated in breeders' liver, which is also the main location of its enzymatic action. The second candidate, *MC2R*, encodes the adrenocorticotropin hormone (ACTH) receptor, which is the main inducer of glucocorticoid synthesis and a crucial component of the hypothalamic–pituitary–adrenal (HPA) axis (*Walker et al., 2015*). In humans and many other mammals, prolonged glucocorticoid excess leads to Cushing's syndrome. Affected individuals exhibit muscle weakness, immune suppression, impairment of the GH/IGF1 axis, higher risk of diabetes, cardiovascular disease (hypertension), osteoporosis, decreased fertility, depression, and weight gain (*Chabre, 2014*; *Ferraù and Korbonits, 2015*). The large overlap of these symptoms with those of aging could explain to some extent that

Cushing's syndrome patients exhibit considerably higher mortality rates (*Etxabe and Vazquez, 1994*). We hypothesized that the increased expression of the ACTH receptor in *Fukomys* non-breeders can cause similar expression patterns and consequences (*Figure 5*).

We subjected this hypothesis (*Figure 5*) to a first test by matching the global fold-changes of the breeder/non-breeder comparison with those of a gene expression comparison of human controls/ Cushing patients (*Hochberg et al., 2015*) and found a significant correlation (R = 0.11, p=6.6*10$^{-36}$). Since the target tissue of Hochberg et al., subcutaneous adipose tissue was unfortunately not included in our study, we used our cross-tissue data for comparison. If we use instead our data from skin as adjacent tissue, a similar correlation results (R = 0.11, p=1.1*10$^{-40}$).

We tested the hypothesis further by checking five of its key predictions. Altered *MC2R* expression (*Figure 6A*) coincides with higher cortisol levels in hair samples from non-breeding *F. mechowii* than in those from breeders of the same species (*Begall et al., 2021*). Furthermore, glucocorticoids such as cortisol exert their effect by binding to the glucocorticoid receptor that, in turn, acts as a transcription factor for many genes (*Gjerstad et al., 2018*). We tested whether the expression of targets of the glucocorticoid receptor (NR3C1) was significantly altered throughout our data using two gene lists (*Phuc Le et al., 2005*): about 300 direct target genes of the receptor that were identified by chromatin immunoprecipitation (i) and about 1300 genes that were found to be differentially expressed depending on the presence or absence of exogenous glucocorticoid (ii). Both gene lists were found to be significantly affected by differential expression at the weighted cross-tissue level (i, p=0.001; ii, p<10$^{-9}$) as well as in five (i) and eight (ii) single tissues (*Supplementary file 1g, h*). In line with our hypothesis, we observed that the weight gain in non-breeders was, on average, twice as strong compared to the weight gain in breeders during the experiment (p=7.49*10$^{-3}$, type II ANOVA, *Figure 6B*). In addition, we found a subtle but significant influence of reproductive status on the density of the vertebrae: the vertebrae of breeders were slightly denser than those of age-matched non-breeders (p=0.03 for vertebra T12 only, and p=0.01 across all examined vertebrae L1, L2, and T12; ANOVA, *Figure 6C*).

## Discussion

The vast lifespan differences between *Fukomys* breeders and non-breeders are, according to our RNA-seq data, associated with only subtle global pattern shifts in transcript levels. Concerning the tested explanatory variables (*Fukomys* species, sex, breeding status), we found by far the highest number of DEGs at the level of the species comparison. Although the number of DEGs between the sexes was comparably high in almost each examined tissue, only very few DEGs were found comparing breeders and non-breeders. One exception is the ovary, whose high number of DEGs corresponds well with the disparity in reproductive activity. Other exceptions are liver, spleen, and, especially, the tissues of the endocrine system (adrenal gland, pituitary gland, thyroid), in which the number of DEGs between breeders and non-breeders ranged from more than 100 to more than 2500.

Changes in the gene expression of the endocrine system are well known to play an important role both in sexual maturation and in aging and the development of aging-associated diseases, for example, diabetes and cardiovascular diseases (*Tatar et al., 2003*; *Chahal and Drake, 2007*; *Jones and Boelaert, 2015*). This finding fits well with the observation of substantial changes in the endocrine system after the transition from non-breeders to breeders in the related naked mole-rat – one of the key results of a recent study in that species (*Bens et al., 2018*). The dominance of differential expression in endocrine tissue is also plausible insofar as these tissues exert a strong control function for other tissues via hormone release.

Steroid hormone biosynthesis exhibits a bipartite pattern in breeders, with up-regulated sex steroid genes and a simultaneous down-regulation of corticosteroid synthesis genes. The former could be expected as a consequence of sexual activity in breeders. For aging, it is, however, interesting that *SULT2A1*, a gene that codes for the specialized sulfotransferase converting the sex steroid DHEA to its non-active form DHEA-S, was found to be heavily down-regulated in breeders. DHEA is the most abundant steroid hormone; it serves as a precursor for sex steroid biosynthesis but also has various metabolic functions on its own (*Allolio and Arlt, 2002*; *Webb et al., 2006*). DHEA levels decrease continuously during the human aging process to an extent that favors it as an aging biomarker (*Maggio, 2007*; *Traish et al., 2011*). As a result, an aggressive advertising of DHEA as 'anti-

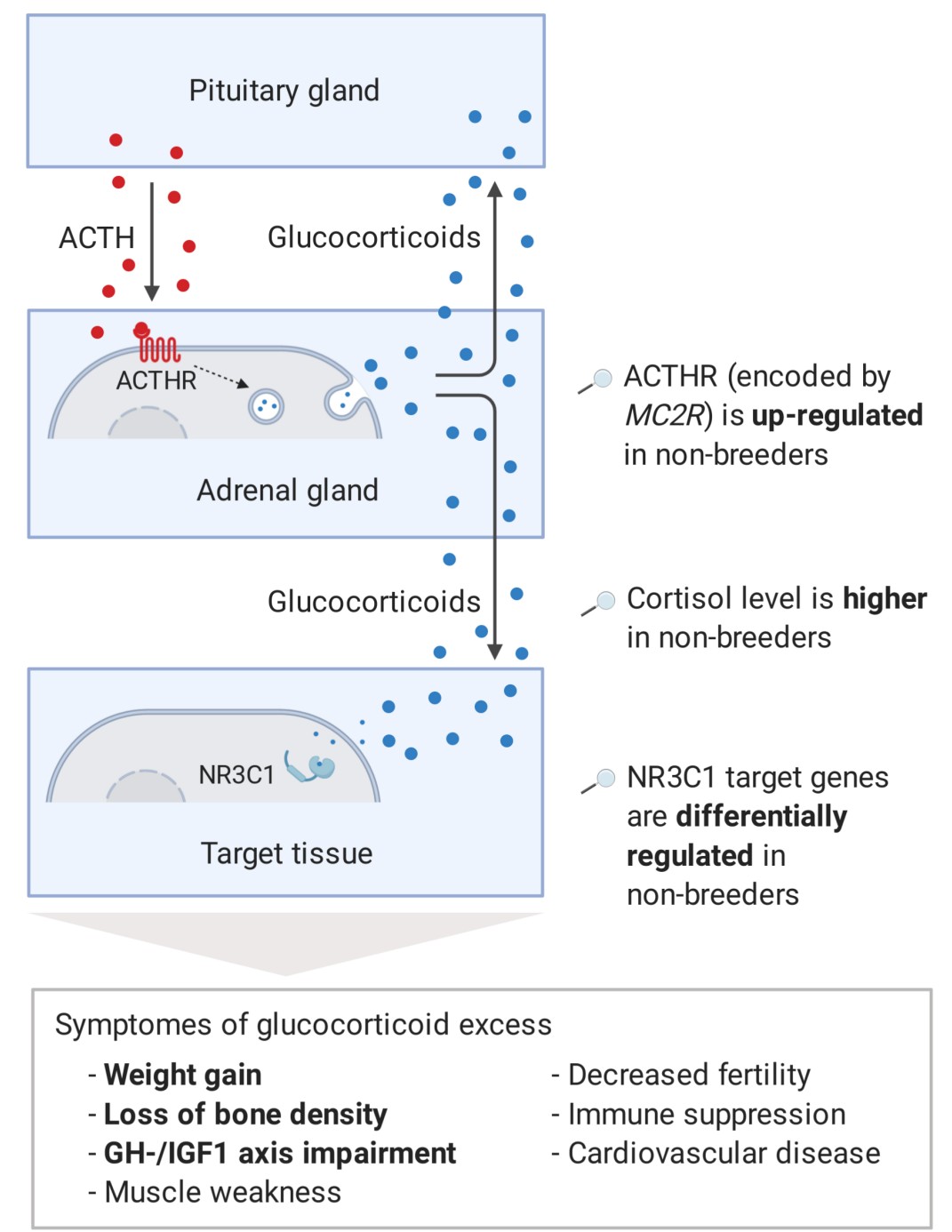

**Figure 5.** Model of the stress axis as a key mechanism for status-dependent lifespan differences in *Fukomys*. From a wide range of mammals, including humans (*Ferraù and Korbonits, 2015*), dogs (*Kooistra and Galac, 2012*), horses (*McCue, 2002*), cats (*Meijer et al., 1978*), and guinea pigs (*Zeugswetter et al., 2007*), it is known that chronic glucocorticoid excess leads to a number of pathological symptoms that largely overlap with those of aging and result in considerably higher mortality rates for affected individuals (*Etxabe and Vazquez, 1994*, #808). The most common cause of chronic glucocorticoid excess is excessive secretion of the adrenocorticotropic hormone (ACTH) by the pituitary gland. ACTH is transported via the blood to the adrenal cortex where it binds to the ACTH-receptor (ACTHR; encoded by the gene *MC2R*), which induces the production of glucocorticoids, especially cortisol. Glucocorticoids are transported to the various tissues, where they exert their effect by activating the glucocorticoid receptor (NR3C1) that acts as a transcription factor and regulates hundreds of genes. The constant overuse of this transcriptional pattern eventually leads to the listed symptoms. Our hypothesis is that the permanent, high expression of the ACTH-receptor in *Fukomys* non-breeders causes effects similar to those known from overproduction of the hormone. In line with this hypothesis, (i) cortisol levels are increased in non-breeders and (ii) target

*Figure 5 continued on next page*

*Figure 5 continued*

genes of the glucocorticoid receptor are highly enriched for status-dependent differential gene expression. Furthermore, the animals were examined for common symptoms of chronic glucocorticoid excess: (iii) non-breeders gained more weight during the experiment than breeders, (iv) exhibited lower bone density at the end of the experiment, and (v) lower gene expression in the growth hormone/insulin-like growth factor 1 axis than breeders.

aging hormone' in the form of dietary supplements could be observed in recent years (*Baulieu, 1996*; *Celec and Stárka, 2003*; *Webb et al., 2006*; *Rutkowski et al., 2014*). Despite conflicting experimental data from various animal studies and clinical trials, a positive effect of DHEA on human health is frequently considered to be likely in the literature. Large-scale and long-term studies, however, are still pending (*Traish et al., 2011*; *Rutkowski et al., 2014*; *Samaras et al., 2015*). Interestingly, DHEA is not only described as an aging marker but also as a marker for clinically relevant glucocorticoid excess (*Burkhardt et al., 2013*).

Regarding corticosteroid synthesis, we hypothesize that the regulation of adrenal gland *MC2R*, coding for the ACTH receptor, causes a considerable proportion of the overall observed expression patterns and of the lifespan extension in breeders (*Figure 5*). As a critical component of the HPA axis, ACTH is a stress hormone that is produced by the pituitary gland and transported by the blood to the adrenal cortex, where it binds to the ACTH receptor (*Fridmanis et al., 2017*). Subsequently, the ACTH receptor induces the synthesis of glucocorticoids (*Walker et al., 2015*), for example, cortisol, which in turn cause immunosuppressive and various metabolic effects throughout the organism (*Becker, 2013*). In many mammals (such as humans, dogs, and guinea pigs) glucocorticoid excess disorder, Cushing's syndrome, is caused by overproduction of the hormone ACTH. Our results for *Fukomys* suggest that the increased levels of the ACTH receptor may lead to symptoms and expression patterns that could resemble some of molecular and phenotypic aspects of this pathological condition (*Figure 5*).

We looked for evidence of a regulation upstream of the ACTH receptor. Given the known positive feedback loop between the ACTH receptor and its own ligand, decreased ACTH synthesis in breeders would have been an obvious explanation (*Imai et al., 2001*). We found that the expression of the ACTH polypeptide precursor gene (*POMC*) is unchanged. However, also other post-

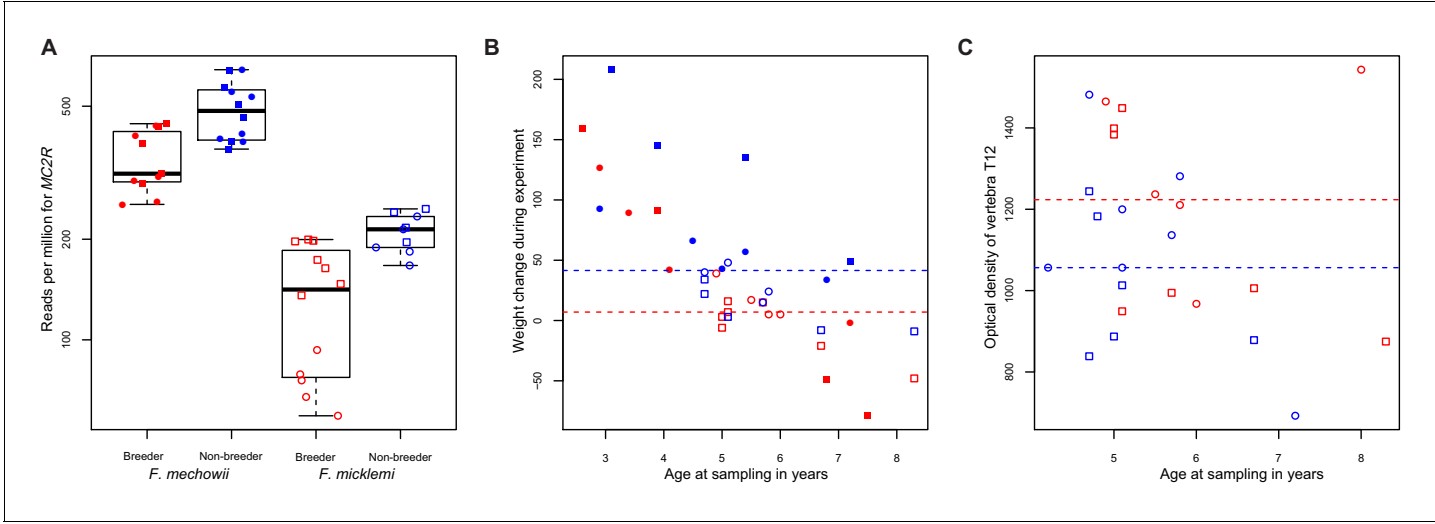

**Figure 6.** *MC2R* gene expression and physiological measurements. (A) Gene expression of *MC2R*, coding for the adrenocorticotropic hormone receptor, in breeders and non-breeders of *Fukomys mechowii* and *Fukomys micklemi*. (B) Weight gain of the animals during the experiment. (C) Measured optical densities of vertebra T12 of *F. micklemi* breeders and non-breeders. Red, breeders; blue, non-breeders; filled, *F. mechowii*; unfilled, *F. micklemi*; circles, females; squares, males; dashed line, median. Statistically significant differences between breeders and non-breeders were determined with (A) DESeq2 (*Love et al., 2014*) and (B, C) analysis of variance with status, species, sex, and age as independent variables (see Materials and methods).

The online version of this article includes the following figure supplement(s) for figure 6:

**Figure supplement 1.** Annual mortality rates in giant mole-rats *Fukomys mechowii*.

transcriptional or post-translational mechanisms such as cleaving may influence the ACTH levels in breeders and non-breeders. Of those genes known to be involved in the regulation of *MC2R* (*Beuschlein et al., 2001*; *Lin et al., 2007*), we found that only *PRKAR1B* was differentially regulated in the adrenal gland. However, this gene codes for only one of seven subunits of the involved protein kinase A. Alternative explanations could be that epigenetic modifications, other still unidentified regulators of the transcript level, or both are responsible for the differential expression of *MC2R*.

While the opportunities to pursue the hypothesis upstream of the ACTH receptor were limited with our data set, many of the predicted downstream effects (*Figure 5*) could be verified. One of the clearest symptoms of Cushing's syndrome known in many animal species is an increase in body fat and a consequent significant weight gain (*Ferraù and Korbonits, 2015*). Our recording of body weights at the beginning and end of the experiment showed that non-breeders gained twice as much weight on average as breeders (p=7.49*10$^{-3}$, type II ANOVA, *Figure 6B*). However, it should be noted that our experimental design does not allow to distinguish between increased body fat content and other possible reasons (i.e., overall growth, muscle gain) for these differences in weight gain. Another known complication associated with glucocorticoid excess is osteoporosis (*Lodish et al., 2010*). A computed tomography measurement of three vertebrae (L1, L2, and T12), in accordance with this, showed significantly lower bone densities of non-breeders (p=0.01, ANOVA, *Figure 6C*). The absolute differences, however, are probably too small to assume that the non-breeders are already physically impaired by that. It seems possible that a longer experiment duration than the 1–2 years we conducted would have aggravated this difference, but this remains speculative. Directly from our gene expression analysis we know that the GH/IGF1 axis of the non-breeders is impaired, which is another well-known symptom of chronic glucocorticoid excess (*Ferraù and Korbonits, 2015*).

In addition, we have systematically compared our observed and expected – based on the Cushing's syndrome hypothesis – expression patterns. A key element in this context is the glucocorticoid receptor (NR3C1), which, when activated by glucocorticoids such as cortisol, is considered to mediate their effects, including those of Cushing's syndrome in the case of chronic activation (*Gjerstad et al., 2018*). According to our hypothesis, the known, experimentally determined NR3C1 target genes (*Phuc Le et al., 2005*) should therefore show a clear pattern of expression differences between breeders and non-breeders. This was the case for both the 1300 indirect (p<10$^{-9}$) and the 300 direct target genes (p=0.001). Furthermore, according to our hypothesis, the expression changes in the comparison of control individuals against patients with Cushing's syndrome should correspond to those of the comparison of breeders vs. non-breeders. A global comparison of fold-changes from human subcutaneous adipose tissue (*Hochberg et al., 2015*) against our cross-tissue data showed a significant correlation (R = 0.11, p=6.6*10$^{-36}$). A very similar result was obtained by comparing the human data with our data from skin as the tissue closest to subcutaneous adipose tissue that was examined in our study (R = 0.11, p=1.1*10$^{-40}$). Despite the clear significance of these correlations, the reported values of the correlation coefficients could be considered to be only modest. It should, however, be taken into account that two evolutionarily widely separated species and not exactly matching tissues were compared.

For further validation, we applied the combination of limma (*Smyth, 2004*) and voom (*Law et al., 2014*) as an alternative DEG detection method to DESeq2 to account for potential dispersion problems in our data. All single DEGs mentioned in the paper were confirmed by this validation. This specifically includes all genes supporting our HPA hypothesis, for example, *SULT2A1* (liver), *MC2R* (adrenal gland), *CYP11A1* (adrenal gland and ovary), *GH* (pituitary gland), and *IGF1* (ovary). In a similar way, we checked the main findings obtained with our standard bioinformatic p-value-based enrichment approach (*Fridley et al., 2010*; *Evangelou et al., 2012*; *Poole et al., 2016*), using a second method that aggregates fold-changes instead. In particular, this step confirmed our initial finding of an enrichment of genes involved in the steroid hormone biosynthesis and ribosome metabolism as a central part of up-regulated anabolism in breeders, consistent with our HPA hypothesis (*Figure 4—figure supplement 11*) (see Control analyses).

In view of the different evidence presented, we consider it justified to assume that the transcriptional, metabolic and physiological condition of *Fukomys* non-breeders resembles that of chronic glucocorticoid excess described from other species. Since this condition shortens life in other species (*Etxabe and Vazquez, 1994*) and several of its symptoms also occur during aging, we hypothesize that it may shorten the lifespans in *Fukomys* non-breeders as well. The ultimate proof and the

quantification of its impact on *Fukomys* lifespan, however, will be difficult in such a long-lived species, which cannot yet be genetically manipulated. Nevertheless, a recent study confirmed one of the most important predictions of the hypothesis: hair cortisol levels of non-breeders are on average about twice as high as those of breeders. However, a subgroup of non-breeders, defined by social environmental factor (one or both parents had died), exhibited cortisol levels nearly as low as in breeders (*Begall et al., 2021*). The drop of cortisol levels in orphaned non-breeders is accompanied by a substantial drop of mortality rates in this subgroup approaching the values of breeders (*Figure 6—figure supplement 1*). This means that across status groups (breeders, orphaned non-breeders, and non-breeders from intact families) mortality rates are clearly correlated with long-term cortisol levels. Taken together, all evidence suggests that the presence of breeders causes distress in non-breeders, resulting in constantly activating the HPA axis and contributing to reduced life expectancy.

In the circulatory system, represented by blood and spleen, down-regulation of coagulation factors was observed in breeders. Because coagulation factors are known to be up-regulated during aging in humans mice, rats, and even fish, this can be interpreted as a sign of a more juvenile breeders' transcriptome (*Ochi et al., 2016*; *Benayoun et al., 2019*). Furthermore, the activity of coagulation factors is associated with a higher risk of coronary heart disease (*Lowe and Rumley, 2014*). However, we found no obvious histopathological lesions in the hearts or other organs (spleen, kidney, liver, lung) of non-breeders. Therefore, if down-regulation of coagulation attenuates the aging process in *Fukomys*, it seems to exert its effect only, if at all, at a latter age not investigated here.

Two defense mechanisms were also found to be up-regulated in breeders: xenobiotic metabolism and TNF-α signaling. Increased TNF-α signaling often leads to the induction of apoptosis (*Annibaldi and Meier, 2018*). In line with this, we found that apoptosis and P53 signaling were also up-regulated in breeders. Apoptosis is considered an important anticancer mechanism (*Baig et al., 2016*; *Pistritto et al., 2016*). We hypothesize this to compensate cancerogenic effects of the anabolic alterations described above (especially the up-regulation of the GH/IGF1 axis). In line with this, our more than 30 years' breeding history with several *Fukomys* species in Germany and the Czech Republic suggests that breeders are as 'cancer-proof' as non-breeders despite their much longer lifespan (own unpublished data and R. Šumbera, personal communication).

Many anabolic pathways are up-regulated in breeders across tissues: protein biosynthesis, myogenesis, and the GH/IGF1 axis. In line with this finding and with the fact that protein synthesis consumes 30–40% of a cell's ATP budget (*Hands et al., 2009*), we observed increased expression of mitochondrial respiratory chain components, implying an increase in the capacity for cellular ATP production. On the other hand, protein degradation and clearance by the proteasome are also up-regulated in breeders. The fact that the expression of proteasomal genes is significantly linked to the genes of ribosome biogenesis and oxidative phosphorylation indicates that those processes influence, or even trigger, each other and hence are regulated in a coordinated manner in *Fukomys* mole-rats.

An important positive regulator of ribosome biogenesis and protein synthesis is mTOR (*Johnson et al., 2013*). Based on this, one could expect an up-regulation of the corresponding gene in our scenario. At the same time, however, the inhibition of mTOR is one of the best-documented life-prolonging interventions from invertebrates to mammals (*Table 2*). Given these conflicting premises, we find mTOR in almost all *Fukomys* tissues as not significantly altered; the exception is the adrenal gland, where mTOR is significantly down-regulated in breeders. It is obvious that, during evolution of *Fukomys* mole-rats, both the extension of breeders' lifespan and an increase in their anabolic processes provided fitness benefits. Consequently, in the underlying organismic framework mTOR may have acquired expression patterns and functions, different from those in organisms studied before. Thus, in the future it may be worth to study *Fukomys* mTOR biology in more detail, particularly in respect to its potential role in naturally evolved ways towards lifespan extension and healthy aging.

The results of differential expression of anabolic components such as the GH/IGF1 axis are surprising. They fall within a debate in aging research that has been highly controversial over time: based on the well-known fact that the expression and secretion of GH and IGF1 decline with age in humans and other mammals (*Bartke, 2019*), *Rudman et al., 1990* administered synthetic GH to elderly subjects in 1990, thereby reversing a number of aging-associated effects such as expansion

**Table 2.** Behavior of important aging-relevant genes and pathways in this study.

| Gene/pathway | Regulation in indicated direction and species can reduce lifespan | Regulation in indicated direction and species can extend lifespan[†] | Differentially expressed in this study in indicated tissues and direction[*] | Gene/pathway is expressed mainly in the following tissues |
|---|---|---|---|---|
| GH1 | Mouse ↑[1] | Mouse ↓[2], rat ↓[2] | Pituitary gland ↑ | Pituitary gland |
| IGF1 | – | Mouse ↓[2] | Adrenal gland ↑, ovary ↑ | Liver, ovary |
| IGF1R | – | Mouse ↓[1], worm ↓[3], fly ↓[3] | Adrenal gland ↓, ovary ↓ | Many |
| KL | Mouse ↓[3] | Mouse ↑[3], worm ↑[3] | Ovary ↓ | Endocrine tissue, kidney |
| SIRT1 | – | Mouse ↑[6], worm ↑[7], fly ↑[7] | – | All |
| MYC | – | Mouse ↓[8] | Thyroid ↑ | All |
| mTOR | – | Mouse ↓[9], worm↓[9], fly ↓[9] | Adrenal gland ↓ | All |
| PRKAA2 (AMPK) | – | Worm ↑[9] | – | Many |
| TP53 | Mouse ↑[11], Fly ↑[11] | Worm ↓[11], mouse ↑[11], fly ↑[11] | – | All |
| SOD2 | – | Worm ↓[10], fly ↑[10] | – | All |
| FOXO3 | Fly ↑[12] | – | Ovary ↓, adrenal gland ↓ | All |
| Protein synthesis | – | Mouse ↓[8], worm ↓[9], fly ↓[9] | Many ↑ | All |
| Proteasome | – | Worm ↑[14], fly ↑[14] | Gonads ↑, adrenal gland ↑ | All |
| Lysosome | – | Worm ↑[13] | – | All |
| Respiratory chain | – | Worm↓[15], fly ↓[16], killifish ↓[17] | Gonads ↑, Adrenal gland ↑, Blood ↑, Spleen ↑ | All |
| Apoptosis | Mouse ↑[11], fly ↑[11] | Worm ↓[11], mouse ↑[11], fly ↑[11] | Skin ↑, pituitary gland ↑ | All |

[*]Direction: breeder/non-breeder.

[†]Direction: old/young.

[1]**Allolio and Arlt, 2002**, [2]**Bartke, 2019**, [3]**van Heemst, 2010**, [4]**Elibol and Kilic, 2018**, [5]**Kwon et al., 2015**, [6]**Satoh et al., 2013**, [7]**Burnett et al., 2011**, [8]**Hofmann et al., 2015**, [9]**Johnson et al., 2013**, [10]**Edrey and Salmon, 2014**, [11]**Feng et al., 2011**, [12]**Giannakou et al., 2004**, [13]**Carmona-Gutierrez et al., 2016**, [14]**Saez and Vilchez, 2014**, [15]**Dillin et al., 2002**, [16]**Copeland et al., 2009**, [17]**Baumgart et al., 2016**.

–: either not affecting lifespan or not known to the best of our knowledge (columns 1 and 2); no change (column 3).

of adipose mass. This led to GH being celebrated as an anti-aging drug (*Junnila et al., 2013*), including dubious commercial offers. Today's aging research, on the contrary, strongly assumes that the enhanced activity of the GH/IGF1 axis accelerates aging and that its suppression could extend lifespan even in humans (*López-Otín et al., 2013*; *Longo et al., 2015*; *Pitt and Kaeberlein, 2015*). In addition to several studies of synthetic GH in humans yielding less convincing results than those of Rudman et al., the main reasons for this turn are the results of studies on short-lived model organisms. From worms to mice, the impairment of the GH/IGF1 axis by genetic intervention consistently led to longer lifespans (*Table 2*), for example, the up-regulation of Klotho – an IGF1 inhibitor – extended the mouse lifespan by as much as 30% (*Kurosu et al., 2005*). As with the impairment of the GH/IGF1 axis, reducing of protein synthesis by decreasing the expression of MYC, a basal transcription factor, extended the mouse lifespan by as much as 20% (*Hofmann et al., 2015*), whereas the impairment of the respiratory chain by rotenone resulted in prolongation of the killifish lifespan by 15% (*Baumgart et al., 2016*; *Table 2*).

Therefore, it is astonishing that massive up-regulation of these anabolic key components accompanies a lifespan extension of approximately 100% in long-lived mammals and potentially even contributes to it. Several points could help to resolve this apparent contradiction: first, the up-regulation of anabolic pathways and key genes is at least partially accompanied by the regulation of other mechanisms that could plausibly compensate for deleterious effects. For example, it is, widely assumed that the negative impact of enhanced protein synthesis on lifespan is to a large extent caused by the accumulation of damaged or misfolded proteins, which is also known to contribute to aging-associated neurodegenerative diseases (*Hipkiss, 2007*; *Saez and Vilchez, 2014*;

*Carmona and Michan, 2016*). Up-regulation of the proteasome, as we observed in breeders in a weighted cross-tissue approach and especially in endocrine tissues, is known to counteract these effects by clearing damaged proteins, leading to lifespan extension in worm and fly (*Saez and Vilchez, 2014*). Enhanced proteasome activity has also been linked with higher longevity of the naked mole-rat compared to the laboratory mouse (*Pérez et al., 2009*; *Rodriguez et al., 2012*) and in comparative approaches across several mammalian lineages (*Pride et al., 2015*). We hypothesize that the simultaneously high anabolic synthesis and catabolic degradation of proteins will lead to a higher protein turnover rate in breeders and, accompanied with that, a reduced accumulation of damaged and misfolded proteins. Similarly, it seems plausible that the up-regulation of the mitochondrial respiratory chain (oxidative phosphorylation) in breeders is compensated for by simultaneous up-regulation of the reactive oxygen hallmark: the mitochondrial respiratory chain is the main source of cellular ROS that can damage DNA, proteins, and other cellular components (*Balaban et al., 2005*; *Starkov, 2008*); the reactive oxygen hallmark consists by definition of genes that are known to be up-regulated in response to ROS treatments. Unsurprisingly, at least 25% of these genes code for typical antioxidant enzymes such as thioredoxin, superoxide dismutase, peroxiredoxin, or catalase that can detoxify ROS. Furthermore, the known cancer-promoting effects of an enhanced GH/IGF1 axis (*Junnila et al., 2013*) could, to some extent, be compensated for by up-regulation of apoptosis and p53 signaling because these are the major mechanisms of cancer suppression (*Bieging et al., 2014*). More generally, potential lifespan-extending effects of moderate up-regulation of both the GH/IGF1 axis and ROS production can also be viewed in the light of the hormesis hypothesis (*Ristow, 2014*), which postulates that mild stressors can induce overall beneficial adaptive stress responses. In line with these arguments, we found higher resting metabolic rates in breeders compared to non-breeders in *F. anselli* (*Schielke et al., 2017*), a species closely related to *F. micklemi*.

A second point that could help to resolve this apparent contradiction concerns the time of intervention. The transition from non-breeder to breeder takes place in adulthood when by far the largest portion of the growth process has already been completed. In contrast, genetic interventions aimed at prolonging the lifespan by inhibiting the GH/IGF1 axis (*Table 2*) often affect the organisms throughout their entire lifespan, including infancy and youth. However, there are some exceptions: for example, late treatments with rapamycin prolong the life of mice almost as much as those that are started early (*Johnson et al., 2013*). Therefore, it is still under debate whether the up-regulation of translation and anabolic processes by the GH/IGF1 axis independently enhances growth and aging or enhances aging only secondarily as a consequence of accelerated growth (*Bartke, 2017*). Our results are an argument for the latter.

A third point is the question of the transferability of knowledge obtained in one species to other species. Most insights into current aging research originate from very short-lived model organisms (*Table 2*). It is clear, on the other hand, that the observed effects of lifespan-prolonging interventions listed in *Table 2* are by far the smallest in the model organisms with the relatively longest lifespans: mice and rats. For example, knockout or knockdown of *IGF1R* in worms that normally live only a few days can result in a lifespan extensions of more than 500% (*Houthoofd et al., 2004*). For mice that have a maximum lifespan of 4 years (*Tacutu et al., 2013*), comparable interventions only lead to an extension of the lifespan by about 25% (*Holzenberger et al., 2003*).It seems reasonable to hypothesize that the effect size of such interventions, regardless of their direction, would diminish further in mole-rats or humans that can live more than 20 or 100 years, respectively, especially if the magnitude of gene expression change is small as in the comparison of breeders and non-breeders. Therefore, it may also be possible that those gene expression patterns caused by our lifespan-extending intervention in *Fukomys* mole-rats highlight particularly the differences in aging mechanisms between short-lived and long-lived species.

As a fourth perspective, one could interpret the down-regulation of the GH/IGF1 axis in non-breeders as a by-product of the apparent up-regulation of the HPA axis in non-breeders, which may well be adaptive in itself. In the wild, non-breeding *Fukomys* mole-rats can maximize their inclusive fitness by either supporting their kin in their natal family or by founding a new family elsewhere. It has therefore been suggested that the shorter lives of non-breeders could be adaptive on the ultimate level if longevity were traded against some other fitness traits, such as competitiveness, to defend colonies against intruders or to enhance their chances for successful dispersal (*Dammann and Burda, 2006*). A constitutively more activated HPA stress axis is expected to offer

advantages for both family defense and dispersal but it carries the risk of status-specific aging symptoms, such as muscle weakness, lower GH/IGF1 axis activity, and lower bone density in the long run (*Ferraù and Korbonits, 2015*). This effect may become even more pronounced under laboratory conditions where grown non-breeders cannot decide to disperse even if they would like to.

However, even the down-regulation of the GH/IGF-axis may be adaptive in itself for non-breeders if it has the potential to protect them from further damage. Note that for today's conventional view that stronger activation of the GH/IGF1 axis accelerates aging (*Table 2*) it is generally challenging that decreasing activity is well documented to correlate with chronological age in a wide range of mammals, including mice, rats, dogs, and humans (*Bartke, 2019*). Also, decreasing activity correlates, under pathological conditions such as Cushing's syndrome, with many symptoms akin to aging (*Figure 5*). It has been suggested that one solution to this apparent contradiction may be that the GH/IGF1 axis is adaptively down-regulated in aging organisms as a reaction to already accumulated aging-related symptoms so that additional damage can be avoided (*Berryman et al., 2008*; *Milman et al., 2016*).

Finally, recent findings of positively selected genes in African mole-rats (family Bathyergidae, containing also *Fukomys* and *Heterocephalus*) could partly explain some of the surprising results. It is striking that translation and oxidative phosphorylation were among the strongest differentially expressed molecular processes concerning the breeding status. Earlier, these processes were also reported to be the most affected by positive selection in the phylogeny of African mole-rats (*Sahm et al., 2018b*). Furthermore, *IGF1* was one of 13 genes that were found to be under positive selection in the last common ancestor of the mole-rats. This may indicate that the corresponding mechanisms were evolutionarily adapted to be less detrimental and make their up-regulation more compatible with a long lifespan. Since the mere fact of positive selection does not permit to draw conclusions about the direction of the mechanistic effect, this hypothesis, however, needs to remain speculative.

## Conclusions

We performed a comprehensive transcriptome analysis that, for the first time within mammalian species, compared naturally occurring cohorts of species with massively diverging longevities. The comparison of faster-aging *Fukomys* non-breeders with similar animals that were experimentally elevated to the slower-aging breeder status revealed by far the most robust transcriptome differences within endocrine tissue: adrenal gland, ovary, thyroid, and pituitary gland. Genes and pathways involved in anabolism, such as *GH*, *IGF1*, translation, and oxidative phosphorylation, were differentially expressed. Their inhibition is among the best-documented life-prolonging interventions in a wide range of short-lived model organisms (*Table 2*). Surprisingly, however, we found that the expression of these mechanisms was consistently higher in slower-aging breeders than in faster-aging non-breeders. This indicates that even basic molecular mechanisms of the aging process known from short-lived species cannot easily be transferred to long-lived species. In particular, this applies to the role of the GH/IGF1 axis, which has in recent years been predominantly described as having a negative impact on lifespan (*López-Otín et al., 2013*; *Pitt and Kaeberlein, 2015*; *Bartke, 2017*). In addition, special features of the mole-rats could also contribute to the explanation of the unexpected result that genes and processes differentially expressed between reproductive statuses were also strongly altered during the evolution of the mole-rats (*Sahm et al., 2018b*). Another intriguing possibility is that, in line with the hormesis hypothesis (*Ristow, 2014*), moderate harmful effects of anabolic processes can be hyper-compensated for by up-regulation of pathways such as proteasomes, P53 signaling, and antioxidant defense against ROS that we observed in slower-aging breeders as well.

Furthermore, our work provides evidence that the HPA stress axis is a key regulator for the observed downstream effects, including the lifespan difference. The effects are likely to be triggered by differential expression of the gene *MC2R* coding for the ACTH receptor, resulting in an altered stress response in breeders vs. non-breeders. This is supported by the facts that (1) cortisol levels in the shorter-lived non-breeders are elevated and (2) life expectancy of non-breeders rises when cortisol levels are reduced. Furthermore, the set of direct and indirect target genes of the glucocorticoid receptors is strongly affected by differential expression, and numerous known downstream effects of glucocorticoid excess have been demonstrated for non-breeders, such as muscle weakness, weight gain, and GH/IGF1 axis impairment. Overall, this evidence suggests that *MC2R* and other genes

along the described signaling pathway are promising targets for possible interventions in aging research.

## Materials and methods

### Animal care and sampling

All animals were housed in glass terraria with dimensions adjusted to the size of the family (min. 40 cm × 60 cm) in the Department of General Zoology, Faculty of Biosciences, University of Duisburg-Essen. The terraria are filled with a 5 cm layer of horticultural peat or sawdust. Tissue paper strips, tubes, and solid shelters were provided as bedding/nesting materials and environmental enrichment. Potatoes and carrots are supplied ad libitum as stable food, supplemented with apples, lettuce, and cereals. *Fukomys* mole-rats do not drink free water. Temperature was kept fairly constant at 26 ± 1°C and humidity at approximately 40%. The daily rhythm was set to 12 hr darkness and 12 hr light.

New breeder pairs (new families) were established between March and May 2014. Each new family was founded by two unfamiliar, randomly chosen adult non-breeders of similar age (min/max/mean: 1.56/6.5/3.58 years in *F. mechowii*; 1.8/5.4/3.1 years in *F. micklemi*) and opposite sex and were taken from already existing separate colonies. These founder animals were moved to a new terrarium in which they were permanently mated. In both species, more than 80% of these new pairs reproduced within the first 12 months (total number of offspring by the end of the year 2016, 82 *F. micklemi* and 81 *F. mechowii*). Only founders with offspring were subsequently assigned to the breeder cohort; founders without offspring were excluded from the study. Non-breeders remained in their natal family together with both parents and other siblings.

*F. mechowii* were sampled in five distinct sampling sessions between March 2015 and winter 2016/2017. *F. micklemi* were sampled in three distinct sampling sessions between November 2016 and July 2017. In both species, females were killed 4–6 months later than their male mates to ensure that these breeder females were neither pregnant nor lactating at the time of sampling in order to exclude additional uncontrolled variables. In total, females/males selected as breeders spent on average 513/353 (*F. mechowii*) and 1095/997 (*F. micklemi*) days in this state before sampling (*Supplementary file 1d, e*).

Before sampling, animals were anesthetized with 6 mg/kg ketamine combined with 2.5 mg/kg xylazine (*Garcia Montero et al., 2015*). Once the animals lost their pedal withdrawal reflex, 1–2 ml of blood was collected by cardiac puncture, and the animals were killed by surgical decapitation. Blood samples (100 µl) were collected in RNAprotect Animal Blood reagent (Qiagen, Venlo, Netherlands). Tissue samples – hippocampus, hypothalamus, pituitary gland, thyroid, heart, skeletal muscle (M. quadriceps femoris), lateral skin, small intestine (ileum), upper colon, spleen, liver, kidney, adrenal gland, testis, and ovary – were transferred to RNAlater (Qiagen, Venlo, Netherlands) immediately after dissection and, following incubation, were stored at −80°C until analysis.

Animal housing and tissue collection were compliant with national and state legislation (breeding allowances 32-2-1180-71/328 and 32-2-11-80-71/345; ethics/animal experimentation approval 84-02.04.2013/A164, Landesamt für Natur-, Umwelt- und Verbraucherschutz Nordrhein-Westfalen).

### RNA preparation and sequencing

For all tissues except blood, RNA was purified with the RNeasy Mini Kit (Qiagen) according to the manufacturer's protocol. Blood RNA was purified with the RNeasy Protect Animal Blood Kit (Qiagen). Kidney and heart samples were treated with proteinase K before extraction as recommended by the manufacturer. Library preparation was performed using the TruSeq RNA v2 kit (Illumina, San Diego, USA), which includes selection of poly-adenylated RNA molecules. RNA-seq was performed by single-end sequencing with 51 cycles in high-output mode on a HiSeq 2500 sequencing system (Illumina) and with at least 20 million reads per sample, as described in *Supplementary file 1i*. We have taken care to concentrate the samples of a tissue on a few sequencing runs and to create a balance of breeders and non-breeders within each run (*Supplementary file 1j*). Read data for *F. mechowii* and *F. micklemi* were deposited as European Nucleotide Archive study with the ID PRJEB29798 (*Supplementary file 1i, j*).

## Read mapping and quantification

It was ensured for all samples that the results of the respective sequencing passed 'per base' and 'per sequence' quality checks of FASTQC (*Andrews, 2020*). The reads were then mapped against previously published and with human gene symbols annotated *F. mechowii* and *F. micklemi* transcriptome data (*Sahm et al., 2018a*; *Sahm et al., 2018b*). For both species, only the longest transcript isoform per gene was used; this is the method of choice for selecting a representative variant in large-scale experiments (*Ezkurdia et al., 2015*; *Source data 1b, c*). This selection resulted in 15,864 reference transcripts (genes) for *F. mechowii* and in 16,400 for *F. micklemi*. After mapping and quantification, we further analyzed only those reference transcripts whose gene symbols were present in the transcript catalogs of both species – this was the case for 15,199 transcripts (the size of the union was 17,065). As mapping, algorithm 'bwa aln' of the Burrows-Wheeler Aligner (*Li and Durbin, 2009*) was used, allowing no gaps and a maximum of two mismatches in the alignment. Only those reads that could be uniquely mapped to the respective gene were used for quantification. Read counts per gene and sample can be found in *Source data 1d*. As another check, we ensured that all samples exhibited a Pearson correlation coefficient of at least 90% in a pairwise comparison based on $log_2$-transformed read counts against all other samples of the same experimental group as defined by samples that were equal in the tissue as well as the species, sex, and reproductive status of the source animal.

## DEGs analysis

p-Values for differential gene expression and fold-changes were determined with DESeq2 (*Love et al., 2014*) and a multifactorial design. The DESeq2 algorithm also includes strict filtering based on a normalized mean gene count that makes further pre-filtering unnecessary (*Love et al., 2014*). Therefore, those genes whose read count was zero for all examined samples were removed before further analysis, thereby reducing the number of analyzed genes to 15,181. The multifactorial design means that, separately for each tissue, we input the read count data of samples across species, sex, and reproductive status into DESeq2 for each sample. This allowed DESeq2 to perform DEG analysis between the two possible states of each of the variables by controlling for additional variance in the other two variables. This approach resulted in a four-times higher sample size than with an approach that would have been based on comparisons of two experimental groups, each of which would be equal in tissue, species, sex, and reproductive status. It is known that the statistical power in RNA-seq experiments can increase considerably with sample size (*Ching et al., 2014*). p-Values were corrected for multiple testing with the Benjamini–Hochberg correction (*Benjamini and Hochberg, 1995*) (FDR).

The results of the DEG analysis can be found in *Source data 1e–g*.

## Enrichment analysis on pathway and cross-tissue level

Let $\left(p_1^t, \ldots p_n^t\right)$ represent the p-values obtained from differential gene expression analysis in the tissue corresponding with index $t$ and the indices $1, \ldots, n$ corresponding to the examined genes. Furthermore, let $\left(p_{x_1}^t, \ldots, p_{x_m}^t\right)$, with $1 \leq x_i \leq n$ and $1 \leq i \leq m$, represent the p-values of genes with the indices $X = (x_1, \ldots, x_m)$ belonging to a corresponding pathway that is tested for enrichment of differential expression signals. To determine the enrichment p-values at the pathway level, we calculated the test statistic $\mathfrak{F}_X^t$ for the gene indices $X$ in tissue index $t$ according to Fisher's method for combining p-values:

$$\mathfrak{F}_X^t = -2 * \sum_{i=1}^{m} \log_e\left(p_{x_i}^t\right)$$

If the underlying test statistics are independent, a combined p-value may be determined by Fishers's method using a $\chi^2$ distribution. However, since this assumption might be violated in gene expression studies (*Väremo et al., 2013*), for example, due to gene regulation cascades, we have empirically estimated the null distributions, as recommended by the literature (*Fridley et al., 2010*; *Evangelou et al., 2012*; *Poole et al., 2016*), using a resampling approach. For this purpose, for each KEGG/MSigDB pathway with m elements, we repeatedly drew $m$ p-values from the total set of p-values $\left(p_1^t, \ldots p_n^t\right)$ and calculated the test statistics. Specifically, this was done by calculating $\mathfrak{F}_{X^j}^t$ for

1000 random draws, each without replacement, $X^j = \left(x_1^j, \ldots, x_m^j\right)$, with $1 \le x_k^j \le n$, $1 \le j \le 1000$. When required for numerical precision, that is, $\le \frac{1}{1000}$, sampling was performed again with 10,000 and 100,000 random draws. The combined p-values for our hypothesis tests can now be obtained by determining the probability of a test statistic in the empirically estimated null distribution. In addition, the indices $X$ were divided into $X_{up}$ and $X_{down}$, depending on whether their fold-change was >1 or <1 in breeder vs. non-breeder comparison, and $\mathfrak{F}_{X_{up}}^t$ and $\mathfrak{F}_{X_{down}}^t$ calculated. The ratio $\frac{\mathfrak{F}_{X_{up}}^t}{\mathfrak{F}_{X_{down}}^t}$ was used as an indicator for functional up- or down-regulation of the corresponding pathway (*Figure 4*, *Figure 4—figure supplements 1* and *2*). Using this approach, enrichment p-values were estimated for all KEGG pathways (*Kanehisa et al., 2017*) and MSigDB hallmarks (*Liberzon et al., 2015*), as well as across all examined tissues (*Source data 2a, b*). In addition, the procedure was applied to test whether the known 300 direct and 1300 indirect glucocorticoid receptor target genes (*Phuc Le et al., 2005*) were enriched for status-dependent differential expression signals (*Supplementary file 1g,h*).

Similarly, cross-tissue DEG-p-values were weighted with a modified test statistic (c.f. *Heard and Rubin-Delanchy, 2018*). The test statistic weights the p-values of the various tissues by the respective expression levels in those tissues. This ensures that, for example, for a ubiquitously expressed gene such as TP53 all tissues contribute relatively equally to the cross-tissue p-value, whereas for typical steroid hormone biosynthesis genes such as CYP11A1 the endocrine tissue results almost exclusively determine the weighted cross-tissue p-value. Given the definitions from above, we calculated the weighted cross-tissue test statistic $\mathcal{F}^g$ for the gene $g$ as follows:

$$\boldsymbol{f}^g = |2 * \sum_{t=1}^{l} \left(log_e\left(p_g^t\right) * w_g^t\right)|$$

with

$$w_g^t = \frac{e\bar{x}pr_g^t * sgn\left(logFC_g^t\right)}{\sum_{t=1}^{l} e\bar{x}pr_g^t}$$

where $logFC_g^t$ is logarithmic fold-change between reproductive states and $e\bar{x}pr_g^t$ is the normalized mean expression (across sexes, species, and reproductive status) for the gene with index $g$ and tissue with index $t$ – both calculated by DESeq2 (*Love et al., 2014*). Furthermore, $sgn$ is the signum function, and $l$ is the number of examined tissues. The test statistic rewards, based on the mentioned assumption, consistency in the direction of gene regulation throughout tissues. All calculated values for $logFC_g^t$, $e\bar{x}pr_g^t$, $p_g^t$, as well as the resulting $\mathcal{F}^g$ and p-values can be found in *Source data 2c*.

Finally, weighted cross-tissue enrichment p-values at the pathway level were estimated by applying the above-described method at the pathway level (based on test statistic $\mathfrak{F}$) to the gene-level weighted cross-tissue p-values. p-Values were corrected for multiple testing with the Benjamini–Hochberg correction (*Benjamini and Hochberg, 1995*) (FDR).

To not solely rely on p-value based statistics, we corroborated the pathways identified as differentially expressed by this procedure on the cross-tissue level by applying an alternative fold-change-based test statistics. The test statistics $\mathfrak{L}_X^t$, for single tissues, and $\mathfrak{l}^g$, for the weighed cross-tissue level, were evaluated by resampling via 1000–100,000 random draws the same way as $\mathfrak{F}_X^t$ and $\mathcal{F}^g$ (see above), to obtain p-values for the examined gene sets:

$$\mathfrak{L}_X^t = \sum_{i=1}^{m} |logFC_{x_i}^t|$$

$$\mathfrak{l}^g = |\sum_{t=1}^{l} \left(logFC_g^t * w_g^t\right)|$$

## Weighted gene co-expression network analysis

We used the WGCNA R package to perform weighted correlation network analysis (*Langfelder and Horvath, 2008*) of all 636 samples at once. We followed the authors' usage recommendation by

choosing a soft power threshold based on scale-free topology and mean connectivity development (we chose power=26 with a soft $R^2$ of 0.92 and a mean connectivity of 38.6), using biweight midcorrelation, setting maxPOutliers to 0.1, and using 'signed' both as network and topological overlap matrix type. The maximum block size was chosen such that the analysis was performed with a single block and the minimum module size was set to 30. The analysis divided the genes into 26 modules, of which 5 were enriched for reproductive status DEGs based on Fisher's exact test and an FDR threshold of 0.05. Those five modules were tested for enrichment among KEGG pathways (*Kanehisa et al., 2017*) with the same test and significance threshold (*Supplementary file 1k*). In addition, module eigengenes were determined and clustered (*Supplementary file 1k*). Then the topological overlap matrix that resulted from the WGCNA analysis ($TOM = \left[ tom_{i,j} \right]$, where the row indices $1 \leq i \leq |examined\ genes|$ correspond to genes and the column indices $1 \leq j \leq |examined\ samples|$ correspond to samples) was used to determine pairwise connectivity between all KEGG pathways that showed differential expression at the weighted cross-tissue level (*Figure 4—figure supplement 3*). Based on the definition of connectivity of genes in a WGCNA analysis (*Langfelder and Horvath, 2008*), we defined the connectivity between two sets of indices $X$ and $Y$ each corresponding to genes as $k_{X,Y} = \sum_{x\ \in X \backslash Y} \sum_{y\ \in Y \backslash X} tom_{x,y}$. p-Values for the connectivities were determined against null distributions that were empirically estimated by determining for each pair $X$ and $Y$ the connectivities of 10,000 pairs of each $|X|$ and $|Y|$ randomly drawn indices (without replacement), respectively. Since 'signed' was used as the network and topological overlap matrix type, the tests were one-sided.

## Other analysis steps

Hierarchical clustering (*Figure 2—figure supplement 1*) was performed based on Pearson correlation coefficients of $\log_2$-transformed read counts between all sample pairs using the complete-linkage method (*Defays, 1977*). The principal variant component analysis (*Figure 2A*) was performed with the pvca package from Bioconductor (*Bushel, 2013*) and a minimum demanded percentile value of the amount of the variabilities, which the selected principal components needs to explain, of 0.5. Enrichments of DEGs among genes enlisted in the Digital Aging Atlas database (*Craig et al., 2015*; *Source data 2d*) were determined with Fisher's exact test, the Benjamini–Hochberg method (FDR) (*Benjamini and Hochberg, 1995*) for multiple test correction, and a significance threshold of 0.05. Pathway visualization (*Figure 4—figure supplements 4–7*) was performed with Pathview (*Luo and Brouwer, 2013*). For the direction analysis of *Fukomys* reproductive status DEGs in previous experiments in naked mole-rats and guinea pigs, we examined those 10 tissues that were examined in all species (*Supplementary file 1f*, *Source data 2e*). Separately for each tissue and combination of species – naked mole-rat or guinea pig – and sex, we determined how many *Fukomys* reproductive status DEGs were up-regulated or down-regulated. We also performed two-sided binomial tests on each of these number pairs with a hypothesized success probability of 0.5. Furthermore, for each combination of species and sex, two-sided exact binomial tests using 0.5 as parameter were performed based on the sums of up-regulated and down-regulated genes across tissues (*Supplementary file 1f*). For enrichment analysis of direct and indirect glucocorticoid receptor target genes, mouse mRNA RefSeq IDs from *Phuc Le et al., 2005* were translated to human Entrez IDs and gene symbols via Ensembl Biomart (*Source data 2f*). To statistically analyze the weight gain of the animals during the experiment, we used a type II ANOVA with status, species, sex, and age as independent variables (*Supplementary file 1b*); the weight gain, defined as the difference in weights at beginning and the end of the experiment, as dependent variable (*Figure 6B*); and no interaction terms. If interaction terms were also used for the model, the p-value for the difference in means between breeders and non-breeders changed from $7.49*10^{-3}$, as reported above, to $7.46*10^{-6}$. To compare our gene expression with Cushing data, we correlated the respective $\log_2$-fold-changes from our skin (as computed by DESeq2) and cross-tissue results with those of human subcutaneous adipose tissue (*Hochberg et al., 2015*). The comparison directions were breeder/non-breeder and control/Cushing patient, respectively. We performed weighted Pearson correlation using the human expression, specifically the baseMean values given in the human study, as weights. Ensembl IDs used in the human study were translated to entrez IDs via the org.Hs.eg.db Bioconductor package (*Source data 2g*).

## Bone density measurements

Frozen carcasses of all *F. micklemi* that had been part of the transcriptome study were scanned with a self-shielded desktop small-animal computed tomography scanner (X-CUBE, Molecubes, Belgium). The x-ray source was a tungsten anode (peak voltage, 50 kVp; tube current, 350 μA; 0.8 mm aluminum filter). The detector was a cesium iodide (CsI) flat panel, building up a screen with $1536 \times 864$ pixels. Measurements were carried for individual 120 ms exposures, with angular sampling intervals of 940 exposures per rotation, for a total of seven rotations and a total exposure time of 789.6 s.

First, we first performed a calibration of the reconstructed CT data in terms of equivalent mineral density. For this purpose, we used a bone density calibration phantom (BDC; QRM GmbH, Moehrendorf, Germany) composed of five cylindrical inserts with a diameter of 5 mm containing various densities of calcium hydroxyapatite (CaHA) surrounded by epoxy resin on a cylindrical shape. The nominal values of CaHA were 0, 100, 200, 400, and 800 mg HA/cm$^3$, corresponding to a density of 1.13, 1.16, 1.25, 1.64, and 1.90 g/cm$^3$ (certified with an accuracy of $\pm0.5\%$). The BDC was imaged and reconstructed with the same specifications as each probe. From the reconstructed Hounsfield units, a linear relationship was determined against the known mineral concentrations.

Reconstruction of the acquired computed tomography data was carried out with an Image Space Reconstruction Algorithm, and spatial resolution was limited to the 100 μm voxel matrix reconstruction. Spherical regions of interest (radius, 0.7 mm) were drawn on the sagittal plane of vertebrae T12, L1, and L2. Care was taken to include all cancellous bone, excluding the cortical edges. Average Hounsfield unit values were computed on the calibration curve to finally retrieve equivalent densities of the regions of interest.

Statistical analysis was performed using general linear models with bone density (Hounsfield units) as dependent variable, age (in days) as continuous covariate, and reproductive status and sex as nominal cofactors. Models were calculated for each vertebra individually (individual models) and across all three vertebrae (full model); in this latter case, vertebral number was added as additional categorial cofactor. In all models, only main effects were calculated, no interactions. Analyses were performed with IBM SPSS version 25 (*Figure 6C*, *Supplementary file 1I*).

## Analysis of mortality rates

Increasing mortality with age is defined as actuarial senescence (e.g., *Møller, 2006*). To calculate status-specific mortality rates across age classes, we subdivided the first 10 years of observation time (corresponding to 1.5–11.5 years of calendar age) into intervals of 6 months. We then recapitulated the times each individual spent as conventional non-breeder (both parents present; cortisol levels high, see *Begall et al., 2021*), orphaned non-breeder (at least one parent absent since more than 6 months; cortisol levels low, see *Begall et al., 2021*), or breeder (cortisol levels low, see *Begall et al., 2021*), and calculated status-specific annual mortality rates for each interval by dividing the number of death events of the respective status during a given interval by the sum of observation years for all individuals holding this status in the same interval. In each interval, individuals that were alive across the entire interval contributed 0.5 years observation time, whereas individuals who died or were censored during that interval contributed the time (expressed in years) up to the death or censorship event. Only animals whose exact age was known were included (i.e., wild-caught animals were excluded). To account for decreasing sample sizes with advancing age, mortality calculations per status were restricted to those intervals containing at least 10 individuals of that status (i.e., around the first 6 years of life for conventional non-breeders and about 11.5 years for orphaned non-breeders and breeders). Status-specific mortality rates were compared by repeated measures ANOVA followed by Tukey post-hoc testing (for the first 6 years) or by paired t-test (breeders vs. all non-breeders, and breeders vs. orphaned non-breeders across all 10 years of observation time) after confirming normal distribution using Shapiro–Wilk normality test using GraphPad Prism.

## Control analyses

For practical reasons, samples collected for this project over several years were sequenced progressively in different batches. Therefore, we analyzed our data and sample scheme across all tissues and both species using BatchQC (*Manimaran et al., 2016*). In summary, the extensive search did not reveal any possible batch effects in the data set from *F. micklemi*, although they may not be ruled out completely for *F. mechowii* (*Source data 2f*). A source for the minor batch effects in *F.*

*mechowii* may be attributable to the repeated sequencing of the same sample(s) in different sequencing experiments – an approach avoided during generation of the *F. micklemi* data set. However, because we followed a robust multifactorial analysis, in which differences must occur consistently across both species to be considered (see DEGs analysis in Materials and methods), false-positive results due to batch effects are substantially reduced (*Manimaran et al., 2016*).

In addition to detecting DEGs with DESeq2 (*Love et al., 2014*), we also examined the central contrast, that is, the juxtaposition of workers and breeders, using the combination of limma (*Smyth, 2004*) and voom (*Law et al., 2014*). The absolute numbers of DEGs found in each tissue, and thus the highly uneven distribution, were nearly identical for all tissues except thyroid (*Figure 2—figure supplement 2*). For the latter, however, a closer look shows that 85% of the DESeq2 DEGs (threshold FDR < 0.05) are confirmed if the FDR threshold for limma/voom is raised from 0.05 to 0.1. If an FDR threshold of 0.05 is applied for both approaches across all tissues, a total of 58% of the DESeq2-DEGs are confirmed by limma/voom. Notably, all genes mentioned in the paper, including the candidates from *Table 1*, are among this set. Using a less strict threshold of 0.1 for limma/voom, 91% of DESeq2-DEGs are confirmed.

We checked the DEG enrichment for potential detection biases with regard to expression level of genes/pathways or size of pathways at the cross-tissue level. For this, we looked at the correlations of the respective p-values with gene expression, pathway expression, and pathway size (*Figure 4—figure supplements 8–10*). In summary, we find a slightly enhanced sensitivity for highly expressed pathways at the cross-tissue level.

In order to avoid potential biases by using solely p-values as indicator of differential gene expression, we additionally resorted to fold-changes as a test statistic for our pathway enrichment analyses. This exercise roughly confirms half of the pathways/hallmarks identified as differentially expressed at the cross-tissue level. Notably the set includes, consistent with the HPA hypothesis, steroid hormone biosynthesis and ribosomes as a central part of up-regulated anabolism in breeders (*Figure 4—figure supplement 11*). Furthermore, myogenesis, the P53 pathway, and coagulation were confirmed as cross-tissue differentially expressed by this approach. On the other hand, proteasome and oxidative phosphorylation gene sets were not identified by the fold-change method.

## Acknowledgements

We thank Ivonne Görlich, Christiane Vole, and Klaus Huse for excellent assistance in the preparation of biological samples. We thank Konstantin Riege for fruitful discussions on the analysis of the data. We thank Debra Weih and Flo Witte for proofreading the manuscript.

## Additional information

### Funding

| Funder | Grant reference number | Author |
| --- | --- | --- |
| Deutsche Forschungsgemeinschaft | PL 173/8-1 | Matthias Platzer |
| Deutsche Forschungsgemeinschaft | DA 992/3-1 | Philip Dammann |
| Deutsche Forschungsgemeinschaft | Research Training Group 1739 | Magdalena Staniszewska |
| Wiedenfeld-Stiftung, Stiftung Krebsforschung Duisburg | | Magdalena Staniszewska |
| Joachim Herz Stiftung | | Arne Sahm |
| Leibniz Association | Open Access Fund | Arne Sahm |

The funders had no role in study design, data collection and interpretation, or the decision to submit the work for publication.

## Author contributions

Arne Sahm, Resources, Data curation, Software, Formal analysis, Validation, Investigation, Visualization, Methodology, Writing - original draft, Project administration, Writing - review and editing; Matthias Platzer, Conceptualization, Resources, Supervision, Funding acquisition, Visualization, Writing - original draft, Writing - review and editing; Philipp Koch, Visualization, Writing - original draft, Writing - review and editing; Yoshiyuki Henning, Marco Groth, Hynek Burda, Paul Van Daele, Resources, Writing - review and editing; Martin Bens, Resources, Formal analysis, Visualization, Writing - review and editing; Sabine Begall, Conceptualization, Resources, Writing - review and editing; Saskia Ting, Moritz Goetz, Magdalena Staniszewska, Jasmin Mona Klose, Pedro Fragoso Costa, Investigation, Writing - review and editing; Steve Hoffmann, Formal analysis, Supervision, Investigation, Methodology, Writing - original draft, Writing - review and editing; Karol Szafranski, Conceptualization, Resources, Formal analysis, Funding acquisition, Investigation, Visualization, Methodology, Writing - original draft, Project administration, Writing - review and editing; Philip Dammann, Conceptualization, Resources, Funding acquisition, Investigation, Visualization, Methodology, Writing - original draft, Project administration, Writing - review and editing

## Author ORCIDs

Arne Sahm https://orcid.org/0000-0002-7330-1790
Matthias Platzer http://orcid.org/0000-0003-0596-8582
Philipp Koch http://orcid.org/0000-0003-2825-7943
Yoshiyuki Henning https://orcid.org/0000-0002-0166-2204
Martin Bens http://orcid.org/0000-0001-7940-6668
Marco Groth http://orcid.org/0000-0002-9199-8990
Sabine Begall http://orcid.org/0000-0001-9907-6387
Karol Szafranski http://orcid.org/0000-0001-6391-1766
Philip Dammann https://orcid.org/0000-0002-0624-3965

## Ethics

Animal experimentation: Animal housing and tissue collection were compliant with national and state legislation (breeding allowances 32-2-1180-71/328 and 32-2-11-80-71/345; ethics/animal experimentation approval 84-02.04.2013/A164, Landesamt für Natur-, Umwelt- und Verbraucherschutz Nordrhein-Westfalen). Before sampling, animals were anaesthetized with ketamine combined with xylazine (Garcia Montero et al. 2015). Every effort was made to minimize suffering.

## Decision letter and Author response

Decision letter https://doi.org/10.7554/eLife.57843.sa1
Author response https://doi.org/10.7554/eLife.57843.sa2

# Additional files

## Supplementary files

• Source data 1. Cross-tissue enrichments details, transcript sequences and DESeq2 results. (**a**) For each Kyoto Encyclopedia of Genes and Genomes pathway and Molecular Signatures Database hallmark that was detected to be significantly (false discovery rate < 0.1) enriched for status-dependent differential expression at the weighted cross-tissue level, the data set contains a *.tsv file with the genes that form the respective pathway/hallmark sorted by their individual contribution to the enrichment. The files also provide an overview of the p-values and fold-changes of those genes in those tissues in which the genes are expressed most highly. (**b**) (fa.gz) Transcript isoforms of *F. mechowii* used for read mapping. (**c**) (fa.gz) Transcript isoforms of *F. micklemi* used for read mapping. (**d**) (tsv.gz) Raw read counts for all 17,065 genes and 636 samples that were analyzed in this study using RNA-seq. (**e**) (zip) DESeq2 results for status-dependent gene expression (direction: breeder/non-breeder). The data set contains one *.tsv file per analyzed tissue. (**f**) (zip) DESeq2 results for sex-dependent gene expression (direction: female/male). The data set contains one *.tsv

file per analyzed tissue. (**g**) (zip) DESeq2 results for species-dependent gene expression (direction: mechowii/micklemi). The data set contains one *.tsv file per analyzed tissue.

• Source data 2. Single-tissue enrichments, cross-tissue gene results, Digital Aging Atlas, comparison with naked mole-rat, glucocorticoid receptor targets, comparison with Cushing's disease and batch effects. (**a**) Enrichment analysis results for status-dependent gene expression on Kyoto Encyclopedia of Genes and Genomes pathways. The data set contains one *.tsv file per analyzed tissue, as well as an additional *.tsv file for the weighted cross-tissue level results. (**b**) (zip) Enrichment analysis results for status-dependent gene expression on Molecular Signatures Database hallmarks. The data set contains one *.tsv file per analyzed tissue, as well as an additional *.tsv file for the weighted cross-tissue level results. (**c**) (tsv.gz) Overview of the weighted cross-tissue differential gene expression analysis. Contains all p-values, fold-changes, and mean expression values for all genes across all tissues as well as the weighted, combined cross-tissue test statistics, p-values, and fold-changes for all genes. (**d**) (tsv) Genes of the Digital Aging Atlas used in this study. (**e**) (zip) Comparison of status-dependent differentially expressed genes with results of an earlier, similar study involving naked mole-rats and guinea pigs. The data set contains one *.tsv file for each of the 10 tissues that were examined in both studies. Each *.tsv file lists the differentially expressed genes for the respective tissue as identified in this study with the determined false discovery rates and fold-changes, as well as the fold-changes determined in the earlier study using naked mole-rats and guinea pigs. (**f**) (zip) Glucocorticoid receptor target genes that were determined by Phuc Le et al. and tested in this study for enrichment of status-dependent differential gene expression. The data set contains one *.tsv file each for target genes determined via chromatin immunoprecipitation and differential gene expression analysis. Phuc Le et al. determined differentially expressed genes between mice that were treated with exogenous glucocorticoids and untreated controls. The relevant table columns from Phuc Le et al. were added by human gene symbol and Entrez IDs that were used for enrichment analysis. (**g**) (tsv) Comparison of Control vs Cushing's disease and breeder vs. non-breeder gene expression. (**h**) (zip) BatchQC reports. This data set contains the results from the analysis of possible batch effects via BatchQC. There is one report for each combination of the 16 tissues and 2 species examined in this study.

• Supplementary file 1. Samples, animals, pairing schemes, comparison with naked mole-rat and guinea pig, glucocorticoid receptor target enrichment, sequencing runs, co-expression networks and bone densities. (**a**) Overview of number of samples that were examined with regard to status, sex, species, and tissue. (**b**) Animal description. (**c**) Overview of the tissues that were successfully sampled for each type of animal. (**d**) Pairing scheme: *F. mechowii*. (**e**) Pairing scheme: *F. micklemi*. (**f**) Analysis of the direction of status-dependent differentially expressed genes that were identified in this study (two *Fukomys* species) in similar experiments with naked mole-rats and guinea pigs (*Bens et al., 2018*, https://www.ncbi.nlm.nih.gov/pmc/articles/PMC6090939/). (**g**) Analysis of status-dependent differential gene expression enrichment on glucocorticoid receptor target genes that were determined by *Phuc Le et al., 2005* (https://www.ncbi.nlm.nih.gov/pmc/articles/PMC1186734/), using chromatin immunoprecipitation. (**h**) Analysis of status-dependent differential gene expression enrichment on glucocorticoid receptor target genes that were determined by *Phuc Le et al., 2005* (https://www.ncbi.nlm.nih.gov/pmc/articles/PMC1186734/), using a differential gene expression analysis. (**i**) Sample descriptions. (**j**) Run descriptions. (**k**) Weighted gene co-expression network analysis module clustering and functional enrichment analysis regarding these modules. (**l**) Bone density measurements.

• Transparent reporting form

## Data availability

Read datasets generated during the current study are available in the European Nucleotide Archive, study ID: PRJEB29798.

The following dataset was generated:

| Author(s) | Year | Dataset title | Dataset URL | Database and Identifier |
|---|---|---|---|---|
| Sahm A, Matthias P, | 2018 | Transcriptome signatures of fast vs. | https://www.ebi.ac.uk/ | European Nucleotide |

| Szafranski K, Dammann P | slow aging in Fukomys mole-rat breeders vs. non-breeders | ena/browser/view/PRJEB29798 | Archive, PRJEB29798 |
|---|---|---|---|

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
