## [Decision Letter]

**Acceptance summary:**

This study analyzed 636 RNA-seq samples across 15 tissues, which is a massive effort and a rich resource. Changes in the hypothalamic-pituitary-adrenal stress axis is found to be associated with the extended life expectancy of reproductive vs. non-reproductive mole-rats, which is consistent with existing knowledge, whereas other associations are contrary to existing knowledge, such as the up-regulation of the IGF1/GH axis and several other anabolic processes in the long lived species. The authors should tune down their claims of causality such as "play a key role", which was not demonstrated without genetic perturbations, by only concluding associations, which are essentially what were described.

**Decision letter after peer review:**

Thank you for submitting your article "The HPA stress axis shapes aging rates in long-lived, social mole-rats" for consideration by *eLife*. Your article has been reviewed by three peer reviewers, and the evaluation has been overseen by a Reviewing Editor and Jessica Tyler as the Senior Editor. The reviewers have opted to remain anonymous.

While we cannot invite revision of the work in its current form, we acknowledge the potential and uniqueness of the data resource you have generated. If in the future you could improve the computational analyses to arrive at more definitive conclusions and include at least limited experimental validation of hypotheses you are advancing, we would be interested to receive such a thoroughly revised submission. We would treat it as a new submission, but we would try to recruit the same reviewers.

Summary:

Differential gene expression between breeding and non-breeding individuals of the two long-lived *Fukomys* species investigated here is a very interesting topic, and the scale of collected data is impressive. Unfortunately, however, both the data analyses and experimental support for the conclusions are rather superficial and descriptive. Given the size and rarity of the dataset, this could be an important resource for the aging research community, if data quality can be clearly demonstrated by robust and extensive computational analyses and experimental validation.

Without demonstrating causality, the authors should refrain from making assertive mechanistic claims, such as "the HPA stress axis shapes aging rates".

Please see the detailed reviews for specific concerns.

*Reviewer #1:*

This is a primarily descriptive study reporting gene expression differences between breeding and non-breeding individuals of two long-lived *Fukomys* species. These mole rat species are interesting for this type of analysis because it has been previously shown that breeders live ~2-fold longer than non-breeders. Thus, it may be possible to learn about mechanisms that determine longevity by studying differences between breeders and non-breeders. Although the manuscript is primarily descriptive, there is value in observational data sets such as this, which can be hypothesis generating and can spur future, more mechanistic studies. I found the presentation to be a bit of a "data dump" with many different comparisons of differentially expressed genes, GO terms, etc., but without a lot in terms true biological insight. As it was largely limited to gene expression, it is also a bit one-dimensional. This work will likely be of interest and value to scientists who study the comparative biology of aging and systems biology of aging, but in its current form seems somewhat incremental and probably will not capture the interest of a broad scientific audience or even the broader gerontology community.

My largest concerns with the content of this manuscript are related to interpretation of the data. While this type of comparative gene expression analysis can generate hypotheses, it cannot strongly support or refute causal relationships without additional experimentation. The authors repeatedly appear to interpret correlative observations with causation and make claims that overreach the data. The title itself is a good example of this where the authors claim "the HPA stress axis shapes aging rates". There is no direct evidence to support this claim or others similar in nature throughout the manuscript.

Another area of overinterpretation is with respect to the relevance of these findings as a test of the validity of prior aging theories or mechanisms. It seems very likely that this is a somewhat unique evolutionary case where upon the switch from non-breeder to breeder there is a dramatic rewiring of physiology at many levels (transcriptional, translational, post-translational, metabolome, epigenome, etc.). Simply because this switch does not match patterns seen in longevity interventions such as caloric restriction, reduced GH/IGF-1 signaling, or mTOR inhibition does not refute or call into question literature describing potential mechanisms for how those interventions act. It likely simply reflects that this is a different path to achieve longevity.

Specific comments:

– The title is problematic as described above.

– The first sentence in the Abstract is problematic as it implies causality between sexual activity/reproduction and longevity. Sexual activity/reproduction are associated with life expectancy. It is interesting to consider whether these two things could perhaps be uncoupled in this animal, as has been shown for reduced fecundity and longevity in invertebrate models.

– The phrase "oppose crucial findings" in the Abstract is also problematic. First, the word "crucial" does not seem to makes sense in this context. What makes them crucial? Second, it is intuitive and expected, indeed perhaps required, that reproduction would be associated with anabolic processes, and this study does not show causality between the observed changes in these processes and longevity, so it does not actually "oppose" the prior findings in genetic models with reduced IGF-1/GH signaling where lifespan is extended.

– I have a problem with the premise that it is possible to "confirm or falsify" results from cross-species or intra-species studies through the type of approach taken here as implied in the text. Confirming or falsifying results implies something about the quality of the prior data itself. I assume what the authors mean is confirming or falsifying the underlying hypotheses or assumptions. Even that is questionable, however, since the mechanisms by which longevity are determined could simply be different within species versus across species versus cases like this where you have dramatically different life expectancies in breeders versus non-breeders.

– I thought the discussion of the GH/IGF-1 results was fairly balanced, but I would encourage the authors to consider more deeply the within species versus across species observations in the literature. The evidence for reduced GH/IGF-1 increasing lifespan comes from within species studies and appears to hold true from worms to dogs and likely in humans as well, although the correlation between body size and lifespan is a bit more complicated in humans for obvious reasons. Within species, smaller individuals who have reduced GH/IGF-1 tend to live longer – that's a correlation. In worms, flies, and mice a reduction in growth signaling has been shown to be sufficient – and very likely causal – for enhanced longevity through genetic and pharmacological studies. The comment that these interventions are all performed during development is not exactly true – in mice rapamycin at least works in adulthood and even when only given transiently or intermittently in adulthood. Across species, larger species tend to live longer. Perhaps the mechanisms going from non-breeders to breeders more resemble the evolutionary longevity strategies that have been taken at the species level rather than the mechanism that appear to determine longevity within species. Personally, I don't see this as a contradiction or a controversy within the field.

– I find the "short-lived" versus "long-lived" species argument to be overly speculative and arbitrary. Interventions such as CR, mTOR inhibition, reduced IGF-1, etc. extend lifespan at least from worms to mice, which is a >50-fold difference in lifespan. Mice to people is ~30-fold. Compared to mice, worms are shorter-lived (by a fold difference metric) than mice are compared to humans.

– The authors state that breeders and non-breeders have "massively diverging aging rates", but I think it is important to keep in mind that differences in lifespan do not necessarily imply differences in aging rate. Especially going from the long-lived state to the short-lived state. Perhaps there is good evidence that functional and molecular declines and diseases/pathologies of aging are accelerated in the non-breeders and delayed in the breeders, which would support this assertion, but this is unclear from the manuscript.

– The phrase "unilaterally described as harmful" in the conclusion is simply not true. GH/IGF-1 signaling limits lifespan in worms, flies, and mice but none of the papers cited unilaterally claim that it is harmful. In fact, some of those same papers note that high GH/IGF-1 signaling often confers a selective advantage in terms of faster maturation and reproduction. So, at the species level, this is beneficial. Even at the individual level, high GH/IGF-1 may be associated with better outcomes in the wild where predation is a factor. High GF/IGF-1 in these species is detrimental (only?) in the context of aging/longevity in a relatively safe laboratory environment at the individual level.

*Reviewer #2:*

Sahm et al. have analyzed gene expression of 22 samples across 15 tissues of *Fukomys* mole rat, and with these resources, they try to explain why breeders and non-breeders have different lifespan in those species. While I was intrigued by the idea and design of the research, I found the bioinformatic approaches, as well as the overall result of the study, fell far short of being acceptable for publication. Below I highlight key aspects of the approach (and related conclusions, or lack thereof) that I believe represent serious issues. One positive note, again, the questions and data of this paper that I believe are highly interesting and important if the author can pursue it in a more focused and solid way.

1) First, in my opinion, the discussion of the paper is not well synthesized, and the content does not help the reader for the aim of study and of what is discussed in the Title (e.g., HPA) and Abstract. While the entire discussion fail to build logics between HPA stress and aging in mole rats, it says story of intervention and hypothesis testing instead, with the last of results immediately jumps into comparisons positive selection genes and DEG (It is unclear why it is relevant even one gene is found under positive selection).

2) Given hundreds of samples have been sequenced in the paper, there is no extensive examination of batch effect, which could ultimately, in its present form, put all the results (e.g., DEG analysis, pathways enrichment, multifactors analysis) and conclusions at high risk. Particularly, the PCA analysis has indicated the species, tissue, and the combination of both variables accounted for 98.4 % of the total variance in the data set, it becomes more important to know how the author have organized the sequencing strategy.

3).Unsophisticated use of enrichment analysis on pathway likely lead to misleading conclusions – a large portion of the analyses presented use an unreasonable approach to identify genes with expression shift across tissues and species to make conclusions that I believe are largely misleading about genes important for longevity. The authors identify pathways that have an overall (gene-wide) changes in gene expression as p value looks like an potential indication of expression shift. While this approach MAY be lucky enough to catch a few of these “true positives", the VAST majority of what it will identify will be pathways with global accumulated changes (which is what the point would be), but rather small pathways or pathways fully of with wired p values or undetectable expression (and thus perhaps some of the least important genes). This overly coarse approach seems to me unreasonable and publishable for this purpose. Then, the extension of these approaches to pathways further muddies the waters of any discussions to the extent that I believe it is largely nonsense that overwhelms the results. The approaches for detecting affected global pathways have been the subject of a very large body of literature, and there are well developed hierarchical/empirical models for testing these hypotheses that the authors have completely and inexcusably ignored. And, of course, it is strong encouraged the author could formulate an new model/index to re-evaluate pathways analysis as the data and experimental design is unique to other study.

4) Last but not the least, small KEGG set, i.e., KEGG that with very few gene could be also confusing such whole categories analysis. The author should check this potential bias using random sampling “pseudo-KEGG set” with same gene number and/or identify a threshold to filer the each true KEGG categories that considered.

*Reviewer #3:*

The manuscript by Sahm et al. describes the transcriptomic comparison of breeders and non-breeders in two species of mole rats from genus Fukomys. The remarkable aspect of *Fukomys* mole rats is that the breeders live significantly longer than workers. The authors produced new breeder couples by pairing animals from different family groups and then compared their transcriptomes to non-breeders of the same age from the original colonies.

There were very few differences identified between breeders and non-breeders. Most transcriptomic changes were confined to gonads and endocrine glands. This is somewhat unsurprising and these organs become active for breeding. The pathways that became activated were related to ribosome biogenesis, protein translation, and MYC signaling, which all reflect physiological activation of these tissues in breeders.

Interestingly, some activation of these signaling pathways was also observed in non-gonadal tissues, which contradicts the common dogma that downregulation of these pathways is associated with lifespan extension. This is a remarkable result that suggests that short-lived model organisms may not correctly reflect signaling effects required for longevity in long-lived species. The authors also point out to higher glucocorticoid levels in workers and speculate that may be showing Cushing's like syndrome.

Overall, this is an important study of high interest.

Items to address:

1) It is not clear how long were the animals maintained in a breeder status prior to analysis.

2) Figure 3A, the rational for the comparison of changes in young breeder versus non breeder animals to changes that occur with age over time is unclear.

3) Figure 4, were the changes driven by gonads mainly? Hoe many were non-gonadal? For example, in Figure 2, muscle showed 0 DEGs by status, but in pathway analysis we see upregulation of pathways. Please explain.

4) Table 2: TOR is downregulated in breeders while ribosome processed are upregulated, please explain.

5) The finding of Cushing's syndrome needs more support. Please consider comparing Cushing's related transcriptome changes as a whole to the changes observed in mole rats, otherwise this conclusion may need to be toned down.

[Editors' note: further revisions were suggested prior to acceptance, as described below.]

Thank you for submitting your article "Slower aging due to sexual activity in mole-rats is associated with transcriptional changes in the HPA stress axis" for consideration by *eLife*. Your article has been reviewed by three peer reviewers, and the evaluation has been overseen by a Reviewing Editor and Jessica Tyler as the Senior Editor. The following individual involved in review of your submission has agreed to reveal their identity: Vera Gorbunova (Reviewer #3).

The reviewers have discussed the reviews with one another and the Reviewing Editor has drafted this decision to help you prepare a revised submission.

We would like to draw your attention to changes in our revision policy that we have made in response to COVID-19 (https://elifesciences.org/articles/57162). Specifically, we are asking editors to accept without delay manuscripts, like yours, that they judge can stand as *eLife* papers without additional data, even if they feel that they would make the manuscript stronger. Thus the revisions requested below only address clarity, data analysis and presentation.

We are glad to see that you smoothed your HPA stress hypothesis according to our suggestions, but further revisions are still necessary to address our previous concerns about data quality and robustness of results have and remove the overinterpretation from the title.

Specifically you need to:

1) Control the batch effect using established packages, such as "BatchQC" or "BEclear";

2) Detect DEG using limma with voom to account for large variances and compare the results with DESeq2's output (see PMID: 23497356);

3) Redo the pathway enrichment with changes of expression values or other static indicators (see PMID: 23625889 and PMID: 32515539 for examples) but not p-value; to remove the bias and validate the HPA hypothesis.

4) Reconsider the title, which is still an over-interpretation of the data and therefore needs to be fixed as follows. The words "slower aging" imply that the same biological aging process is occurring in the breeders and non-breeders, which is not actually clear from the data. Indeed, it seems quite plausible that the non-breeders die prematurely due to too much stress (as you speculate). Whether the rate of aging is different or not remains unclear. Replacing "slower aging" with "Increased longevity" or something like that would address this issue.

---

## [Author Response]

Reviewer #1:This is a primarily descriptive study reporting gene expression differences between breeding and non-breeding individuals of two long-lived Fukomys species. These mole rat species are interesting for this type of analysis because it has been previously shown that breeders live ~2-fold longer than non-breeders. Thus, it may be possible to learn about mechanisms that determine longevity by studying differences between breeders and non-breeders. Although the manuscript is primarily descriptive, there is value in observational data sets such as this, which can be hypothesis generating and can spur future, more mechanistic studies. I found the presentation to be a bit of a "data dump" with many different comparisons of differentially expressed genes, GO terms, etc., but without a lot in terms true biological insight. As it was largely limited to gene expression, it is also a bit one-dimensional. This work will likely be of interest and value to scientists who study the comparative biology of aging and systems biology of aging, but in its current form seems somewhat incremental and probably will not capture the interest of a broad scientific audience or even the broader gerontology community.My largest concerns with the content of this manuscript are related to interpretation of the data. While this type of comparative gene expression analysis can generate hypotheses, it cannot strongly support or refute causal relationships without additional experimentation. The authors repeatedly appear to interpret correlative observations with causation and make claims that overreach the data. The title itself is a good example of this where the authors claim "the HPA stress axis shapes aging rates". There is no direct evidence to support this claim or others similar in nature throughout the manuscript.Another area of overinterpretation is with respect to the relevance of these findings as a test of the validity of prior aging theories or mechanisms. It seems very likely that this is a somewhat unique evolutionary case where upon the switch from non-breeder to breeder there is a dramatic rewiring of physiology at many levels (transcriptional, translational, post-translational, metabolome, epigenome, etc.). Simply because this switch does not match patterns seen in longevity interventions such as caloric restriction, reduced GH/IGF-1 signaling, or mTOR inhibition does not refute or call into question literature describing potential mechanisms for how those interventions act. It likely simply reflects that this is a different path to achieve longevity.

We thank you for your outright comment that we have made too strong statements in light of the evidence presented. Your review was the first we received for this manuscript, and considerably helped us to realize the pitfalls of over-interpretation. We implemented your constructive criticism in two ways. First, we have, in accordance with your suggestions, toned down or deleted statements and generally made the correlative character of the facts presented more transparent. In addition, we are now discussing the previously collected and new, complementary evidence for the HPA stress hypothesis in more detail and with more differentiation than before.

Specific comments:– The title is problematic as described above.

We agree and use a new title that reflects the correlative character of the underlying evidence: Slower aging due to sexual activity in mole-rats is associated with transcriptional changes in the HPA stress axis

– The first sentence in the Abstract is problematic as it implies causality between sexual activity/reproduction and longevity. Sexual activity/reproduction are associated with life expectancy. It is interesting to consider whether these two things could perhaps be uncoupled in this animal, as has been shown for reduced fecundity and longevity in invertebrate models.

Yes, we modified the Abstract according to your suggestions. Specifically, we made the following replacements:

“Sexual activity and/or reproduction are associated with a doubling of life expectancy in the long-lived rodent genus *Fukomys*.”

“This analysis suggests that changes in the regulation of the hypothalamic-pituitary-adrenal stress axis play a key role regarding the extended life expectancy of reproductive vs. non-reproductive mole-rats. This is substantiated by a corpus of independent evidence.”

– The phrase "oppose crucial findings" in the Abstract is also problematic. First, the word "crucial" does not seem to makes sense in this context. What makes them crucial? Second, it is intuitive and expected, indeed perhaps required, that reproduction would be associated with anabolic processes, and this study does not show causality between the observed changes in these processes and longevity, so it does not actually "oppose" the prior findings in genetic models with reduced IGF-1/GH signaling where lifespan is extended.

We agree and modified the Abstract according to your suggestions:

“On the other hand, several of our results are not consistent with knowledge about aging of short-lived model organisms.”

Moreover, we added to the Introduction:

“Nevertheless, both *Fukomys* and naked mole-rats can be seen as counter-examples to the disposable soma theory of aging, which assumes that the aging process is linked to an evolutionary tradeoff in which the preservation of the organism is essentially sacrificed to reproductive success (Kirkwood, 1977).”

– I have a problem with the premise that it is possible to "confirm or falsify" results from cross-species or intra-species studies through the type of approach taken here as implied in the text. Confirming or falsifying results implies something about the quality of the prior data itself. I assume what the authors mean is confirming or falsifying the underlying hypotheses or assumptions. Even that is questionable, however, since the mechanisms by which longevity are determined could simply be different within species versus across species versus cases like this where you have dramatically different life expectancies in breeders versus non-breeders.

Yes, we agree and removed the respective lines from the manuscript.

– I thought the discussion of the GH/IGF-1 results was fairly balanced, but I would encourage the authors to consider more deeply the within species versus across species observations in the literature. The evidence for reduced GH/IGF-1 increasing lifespan comes from within species studies and appears to hold true from worms to dogs and likely in humans as well, although the correlation between body size and lifespan is a bit more complicated in humans for obvious reasons. Within species, smaller individuals who have reduced GH/IGF-1 tend to live longer – that's a correlation. In worms, flies, and mice a reduction in growth signaling has been shown to be sufficient – and very likely causal – for enhanced longevity through genetic and pharmacological studies.

That is, if we see it correctly, so far completely in line with the manuscript, which describes that today’s literature based on evidence in different species predominantly assumes that increased IGF1/GH signaling shortens life and decreased signaling prolongs it (Table2). If we have misunderstood you on this point and in your opinion important arguments or literature evidence is still missing, we would be happy to add it.

The comment that these interventions are all performed during development is not exactly true – in mice rapamycin at least works in adulthood and even when only given transiently or intermittently in adulthood.

Yes, we have supplemented and qualified the statement as follows:

“However, there are some exceptions: for example, late treatments with rapamycin prolong the life of mice almost as much as those that are started early (Johnson et al., 2013). Therefore, it is still under debate…”

Across species, larger species tend to live longer. Perhaps the mechanisms going from non-breeders to breeders more resemble the evolutionary longevity strategies that have been taken at the species level rather than the mechanism that appear to determine longevity within species. Personally, I don't see this as a contradiction or a controversy within the field.

We understand your concerns, however, here we are making the case that there is no need to postulate two general strategies/mechanisms for longevity determination at the inter- and intraspecies levels. For example, in both intra- and interspecies comparisons, the growth rate is clearly anticorrelated with lifespan (https://pubmed.ncbi.nlm.nih.gov/11868658/, https://www.ncbi.nlm.nih.gov/pmc/articles/PMC3574304/, https://pubmed.ncbi.nlm.nih.gov/12954479/). In respect to the well accepted positive correlation of body weight and longevity this means, that large long-lived species grow relative longer than small short-lived ones. Noteworthy, in mammals maximum life span is even stronger correlated to time to maturity than either to adult weight or growth rate (see, e.g., Figure 2 of https://www.ncbi.nlm.nih.gov/pmc/articles/PMC4406664/). In respect to the manuscript, however, we believe that such a speculation and discussion would extend too far from the topics of our work.

– I find the "short-lived" versus "long-lived" species argument to be overly speculative and arbitrary. Interventions such as CR, mTOR inhibition, reduced IGF-1, etc. extend lifespan at least from worms to mice, which is a >50-fold difference in lifespan. Mice to people is ~30-fold. Compared to mice, worms are shorter-lived (by a fold difference metric) than mice are compared to humans.

We would like to point out that the concept of comparing short- versus long-lived has been applied successfully (see e.g. Austad SN, Comparative biology of aging, PMID: 19223603 and Gorbunova V, Comparative genetics of longevity and cancer, PMID: 24981598). Particular in species with a similar biology like mammals, where a strong correlation exists between body-weight and longevity, it is quite rational and powerful. Based on this correlation, short- and long-lived outliers can be objectively identified. Classification of *Fukomys* as a long-lived outlier is a basic motivation of our research efforts.

However, we understand your concern with respect to the fact that the mentioned treatments work in principle consistently in the same direction in a wide range of species. Therefore, we have shortened the paragraph considerably and concentrated on what we consider a more stringent argument:

“A third point is the question of the transferability of knowledge obtained in one species to other species. Most insights into current aging research originate from very short-lived model organisms (Table 2). […]Therefore, it may also be possible that those gene expression patterns caused by our lifespan-extending intervention in *Fukomys* mole-rats highlight particularly the differences in aging mechanisms between short-lived and long-lived species.”

– The authors state that breeders and non-breeders have "massively diverging aging rates", but I think it is important to keep in mind that differences in lifespan do not necessarily imply differences in aging rate. Especially going from the long-lived state to the short-lived state. Perhaps there is good evidence that functional and molecular declines and diseases/pathologies of aging are accelerated in the non-breeders and delayed in the breeders, which would support this assertion, but this is unclear from the manuscript.

Sorry for being insufficiently clear in this respect in the submitted manuscript. We agree of course that differences in lifespan do not necessarily imply differences in aging rates, as organisms can die for various reasons, many of which have little to do with physiological aging, and as a matter of fact, we have not actually measured aging rates in our study species. However, the survival data for *Fukomys* non-breeders and breeders (https://pubmed.ncbi.nlm.nih.gov/16488857/, https://www.ncbi.nlm.nih.gov/pmc/articles/PMC3076438/) have been obtained in protected captive environments where both groups lived essentially under the same conditions, and where starvation, predation or dispersal could be excluded as mortality causes. Accidental death cases were very rare. Kaplan-Meier-survival curves and analyses of mortality rates indicate that both groups show actuarial senescence, but with obviously earlier onset of aging and significantly higher annual mortality hazards in non-breeders across all age classes (https://pubmed.ncbi.nlm.nih.gov/16488857/, https://www.ncbi.nlm.nih.gov/pmc/articles/PMC3076438/, https://cloud.leibniz-fli.de/index.php/s/d2Wob9jFAeyRHQZ [accepted at Philos. Trans. R. Soc. B]). Hence, we are confident to say that the longer mean and absolute lifespans of breeders result from a combination of delayed onset of aging and henceforth lower age-specific mortality rates in *Fukomys* breeders as compared to non-breeders. As, in the absence of extrinsic mortality causes, both parameters are parameters of aging, we consider it justified to speak of “slower aging breeders” and/or “faster aging non-breeders” whenever appropriate.

However, we modified the text to avoid misunderstandings:

“… compared naturally occurring cohorts of species with massively diverging longevities.”

plus numerous other passages where the phrase “aging-rate” has been substituted likewise.

– The phrase "unilaterally described as harmful" in the conclusion is simply not true. GH/IGF-1 signaling limits lifespan in worms, flies, and mice but none of the papers cited unilaterally claim that it is harmful. In fact, some of those same papers note that high GH/IGF-1 signaling often confers a selective advantage in terms of faster maturation and reproduction. So, at the species level, this is beneficial. Even at the individual level, high GH/IGF-1 may be associated with better outcomes in the wild where predation is a factor. High GF/IGF-1 in these species is detrimental (only?) in the context of aging/longevity in a relatively safe laboratory environment at the individual level.

Yes, we phrased that wrongly. In our discussion, we even deal with the possible evolutionary advantages of increased IGF1/GH signalling in non-breeders in such a tradeoff view.

“In particular, this applies to the role of the GH/IGF1 axis, which has in recent years been predominantly described as having a negative impact on lifespan (Lopez-Otin et al., 2013; Pitt and Kaeberlein, 2015; Bartke, 2017).”

Reviewer #2:Sahm et al. have analyzed gene expression of 22 samples across 15 tissues of Fukomys mole rat, and with these resources, they try to explain why breeders and non-breeders have different lifespan in those species. While I was intrigued by the idea and design of the research, I found the bioinformatic approaches, as well as the overall result of the study, fell far short of being acceptable for publication. Below I highlight key aspects of the approach (and related conclusions, or lack thereof) that I believe represent serious issues. One positive note, again, the questions and data of this paper that I believe are highly interesting and important if the author can pursue it in a more focused and solid way.1) First, in my opinion, the discussion of the paper is not well synthesized, and the content does not help the reader for the aim of study and of what is discussed in the Title (e.g., HPA) and Abstract. While the entire discussion fail to build logics between HPA stress and aging in mole rats, it says story of intervention and hypothesis testing instead, with the last of results immediately jumps into comparisons positive selection genes and DEG (It is unclear why it is relevant even one gene is found under positive selection).

In response to the comment, we now discuss the hypotheses, results and verifications concerning the HPA axis in greater detail. Furthermore, we have shortened those parts of the manuscript describing the up-regulation of various anabolic components such as the IGF1/GH axis, which are predominantly associated with a shortening of lifespan (see answer 1.10 to Reviewer #1). You also assert that we jump “immediately” into a comparison of positively selected genes and DEGs. This is very surprising to us, since this issue is addressed only once in the entire manuscript: in the last of about fifteen paragraphs of our Discussion – by far not the main pillar of our manuscript.

2) Given hundreds of samples have been sequenced in the paper, there is no extensive examination of batch effect, which could ultimately, in its present form, put all the results (e.g., DEG analysis, pathways enrichment, multifactors analysis) and conclusions at high risk. Particularly, the PCA analysis has indicated the species, tissue, and the combination of both variables accounted for 98.4 % of the total variance in the data set, it becomes more important to know how the author have organized the sequencing strategy.

We have added the rational one for our sequencing strategy as well as a new supplement table that details the organization of sequencing experiments together. In brief, we have taken great care to distribute samples of breeders and non-breeders across the sequencing runs to avoid such systematic errors.

“ We have taken care to concentrate the samples of a tissue on a few sequencing runs and to create a balance of breeders and non-breeders within each run (Supplementary file 1J).”

3) Unsophisticated use of enrichment analysis on pathway likely lead to misleading conclusions – a large portion of the analyses presented use an unreasonable approach to identify genes with expression shift across tissues and species to make conclusions that I believe are largely misleading about genes important for longevity. The authors identify pathways that have an overall (gene-wide) changes in gene expression as p value looks like an potential indication of expression shift. While this approach MAY be lucky enough to catch a few of these “true positives", the VAST majority of what it will identify will be pathways with global accumulated changes (which is what the point would be), but rather small pathways or pathways fully of with wired p values or undetectable expression (and thus perhaps some of the least important genes). This overly coarse approach seems to me unreasonable and publishable for this purpose. Then, the extension of these approaches to pathways further muddies the waters of any discussions to the extent that I believe it is largely nonsense that overwhelms the results. The approaches for detecting affected global pathways have been the subject of a very large body of literature, and there are well developed hierarchical/empirical models for testing these hypotheses that the authors have completely and inexcusably ignored. And, of course, it is strong encouraged the author could formulate an new model/index to re-evaluate pathways analysis as the data and experimental design is unique to other study.ℱ=−2*∑i=1mloge(pi)

Since we have reason to believe that this could be a methodological misunderstanding (see below), we would like to take the opportunity to explain the basic aspects of our approach again with references to the relevant literature. In order to combine different p-values in a threshold-free way into a test statistic, we have used the Fisher’s method.In fact, Fisher's method is one of the most widely used methods in statistics (https://academic.oup.com/biomet/article/105/1/239/4788722). The provided reference also illustrates the use of weights. This is an appropriate approach to account for different levels of gene expression in the individual tissues in the context of cross-tissue analysis. In addition, the method is also specifically recommended for the identification of differentially expressed pathways in gene expression analyses (e.g. https://journals.plos.org/plosone/article?id=10.1371/journal.pone.0012693,

https://academic.oup.com/biomet/article/105/1/239/4788722, https://journals.plos.org/plosone/article?id=10.1371/journal.pone.0041018, https://www.tandfonline.com/doi/abs/10.1080/19466315.2014.888013, https://bmcbioinformatics.biomedcentral.com/articles/10.1186/1471-2105-14-368, https://bmcgenomics.biomedcentral.com/articles/10.1186/1471-2164-8-96, https://www.ncbi.nlm.nih.gov/pmc/articles/PMC3632109/).

“Simulation results demonstrated that overall Fisher's method and the global model with random effects have the highest power for a wide range of scenarios” (first linked article)

Unfortunately, you did not specify any instance of the "large body of literature" that you consider better suited for threshold-free, weighted enrichment analyses of gene expression comparisons across several tissues.

Strategies used to determine the combined p-value based on Fisher's method, are described, for example, in the first, third and seventh linked article. In general Fisher’s method assumes independence of m individual test statistics and uniformity of the associated p-values under the null. Consequently, the test statistic obtained with Fisher’s method is approximately chi square distributed with 2m degrees of freedom. In gene expression analyses, however, the assumption of independence may be violated (https://journals.plos.org/plosone/article?id=10.1371/journal.pone.0012693, https://www.ncbi.nlm.nih.gov/pmc/articles/PMC3632109/). Therefore, a Monte-Carlo approach is often used instead to determine the p-values with empirically determined null distributions.

We have taken your comments as an opportunity to describe the procedure in more detail:

“If the underlying test statistics are independent, a combined p-value may be determined by Fishers’s method using an χ^2^-distribution. […] The combined p-values for our hypothesis tests can now be obtained by determining the probability of a test statistic in the empirically estimated null-distribution.”

Regarding the assertions about validity and integrity of our results we would like to refer to our cross-tissue-level analysis. We are able to falsify the hypothesis that our method preferentially identifies poorly expressed genes/pathways or small pathways:

“We checked whether this method may have a detection bias with regard to expression level of genes/pathways or size of pathways at the cross-tissue level. For this, we looked at the correlations of the respective p-values with gene expression, pathway expression, and pathway size (Figure 4—figure supplement 8-10). In summary, we find a slightly enhanced sensitivity for highly expressed pathways at the cross-tissue level.”

Furthermore, a preference for pathways in terms of size can already be excluded on the basis of the empirical determination of the null distribution described above. There, the number of elements in the respective pathway is taken into account (see above).

4) Last but not the least, small KEGG set, i.e., KEGG that with very few gene could be also confusing such whole categories analysis. The author should check this potential bias using random sampling “pseudo-KEGG set” with same gene number and/or identify a threshold to filer the each true KEGG categories that considered.

Confusingly, the description of this *“pseudo-KEGG set”* approach, which you suggest as a possible solution, matches exactly the method used in our manuscript. Given this last statement, we believe that the fundamental criticism our method may have been the result of a misunderstanding. We sincerely hope to have resolved this with our explanations.

Reviewer #3:The manuscript by Sahm et al. describes the transcriptomic comparison of breeders and non-breeders in two species of mole rats from genus Fukomys. The remarkable aspect of Fukomys mole rats is that the breeders live significantly longer than workers. The authors produced new breeder couples by pairing animals from different family groups and then compared their transcriptomes to non-breeders of the same age from the original colonies.There were very few differences identified between breeders and non-breeders. Most transcriptomic changes were confined to gonads and endocrine glands. This is somewhat unsurprising and these organs become active for breeding. The pathways that became activated were related to ribosome biogenesis, protein translation, and MYC signaling, which all reflect physiological activation of these tissues in breeders.Interestingly, some activation of these signaling pathways was also observed in non-gonadal tissues, which contradicts the common dogma that downregulation of these pathways is associated with lifespan extension. This is a remarkable result that suggests that short-lived model organisms may not correctly reflect signaling effects required for longevity in long-lived species. The authors also point out to higher glucocorticoid levels in workers and speculate that may be showing Cushing's like syndrome.Overall, this is an important study of high interest.Items to address:1) It is not clear how long were the animals maintained in a breeder status prior to analysis.

Yes, we added the detailed information for every animal in the study to the supplement and summarized it in the manuscript as follows:

“In total, females/males selected as breeders spent on average 513/353 (*F. mechowii*) and 1095/997 (*F. micklemi*) days in this state before sampling (Supplementary file 1D,E).”

2) Figure 3A, the rational for the comparison of changes in young breeder versus non breeder animals to changes that occur with age over time is unclear.

We thank you for the question and introduced the two corresponding paragraphs with the rationale:

“To gain insights into the molecular basis of the bimodal aging pattern in *Fukomys,* it would be of course interesting to compare sufficiently large cohorts of old breeders and non-breeders both at similar chronological and biological ages. But as old animals, particularly of a long-lived species, are extremely valuable due to obvious logistical, timely and financial reasons, such an approach was impossible for us. As an alternative, we decided to study young breeders and non-breeders, that are much easier to obtain, and hypothesized that the status change marks the beginning of a slowdown in the aging process.”

“Since the change from non-breeder to breeder status apparently marks the beginning of a slowdown in the aging process, we first wanted to find out whether and where there are intersections of reproductive status DEGs with those whose expression level is known to change during aging. Therefore, we determined overlaps by using…”

3) Figure 4, were the changes driven by gonads mainly? Hoe many were non-gonadal? For example, in Figure 2, muscle showed 0 DEGs by status, but in pathway analysis we see upregulation of pathways. Please explain.

The enrichment analysis is not based on the differentially expressed genes only but was performed in a threshold-free manner. This was done in order not to lose the ability to detect differences at the pathway level for which the statistical power at the single gene level is insufficient due to the available sample size. An enrichment analysis that would focus only on DEGs weights all genes with a p-value below a certain threshold (e.g. 0.05) with 1 and all genes above this threshold with 0. Instead, we have adopted an approach that takes each gene of a pathway into account in the test statistics – weighted according to its p-value, providing test sensitivity in tissues with no or few detected DEGs. We have addressed this approach in the Results section of our manuscript as follows:

“For this, we used Kyoto Encyclopedia of Genes and Genomes (KEGG) pathways (Kanehisa et al., 2017), and Molecular Signatures Database (MSigDB) hallmarks (Liberzon et al., 2015) as concise knowledge bases. We used an established method that combines all p-values of genes in a given pathway in a threshold-free manner. The advantage of this approach is that it bundles the p-values from test results of individual gene expression differences at the level of pathways (see Materials and methods section for details).”

To avoid confusion when viewing the figure in isolation or when the paragraph above has been overlooked, we have also added the following to the figure description:

“This enrichment analysis was carried out threshold-free, which is why tissues without differentially expressed individual genes (see Figure 2B) can also show differentially expressed pathways.”

4) Table 2: TOR is downregulated in breeders while ribosome processed are upregulated, please explain.

It should be noted that the mTOR downregulation, as shown in Table 2, is limited to the adrenal gland only. Nevertheless, some interesting questions do indeed arise with regard to mTOR regulation due to the scenario on the one hand (massive life span differences) and the concrete results on the other hand (mTOR largely unchanged, but protein biosynthesis up-regulated). Although we cannot conclusively answer these questions, we will discuss them in a new paragraph:

“An important positive regulator of ribosome biogenesis and protein synthesis is mTOR (Johnson et al., 2013). Based on this, one could expect an upregulation of the corresponding gene in our scenario. At the same time, however, the inhibition of mTOR is one of the best documented life-prolonging interventions from invertebrates to mammals (Table 2). Given these conflicting premises, we find mTOR in almost all *Fukomys* tissues as not significantly altered; the exception is the adrenal gland, where mTOR is significantly downregulated in breeders. It is obvious that, during evolution of *Fukomys* mole-rats, both the extension of breeders’ lifespan and an increase in their anabolic processes provided fitness benefits. Consequently, in the underlying organismic framework mTOR may have acquired expression patterns and functions, different from those in organisms studied before. Thus, in the future it may be worth to study *Fukomys* mTOR biology in more detail, particularly in respect to its potential role in naturally evolved ways towards lifespan extension and healthy aging.”

5) The finding of Cushing's syndrome needs more support. Please consider comparing Cushing's related transcriptome changes as a whole to the changes observed in mole rats, otherwise this conclusion may need to be toned down.

Many thanks for this really good suggestion! We were able to find one single data set that performs an RNA-seq comparison of Cushing vs. controls. We have correlated this data on a global level with our non-breeder/breeder comparison and have extended the Materials and methods and the last section of the Results accordingly. We discuss the new results and the limitations of this comparison together with our other hypothesis tests about the HPA axis and Cushing in a new section. The part corresponding to your suggestion is this:

“Furthermore, according to our hypothesis, the expression changes in the comparison of control individuals against patients with Cushing's syndrome should correspond to those of the comparison of breeders vs. non-breeders. A global comparison of fold-changes from human subcutaneous adipose tissue (Hochberg et al., 2015) against our cross-tissue data showed a significant correlation (R=0.11, p=6.6*10^-36^). A very similar result was obtained by comparing the human data with our data from skin as the tissue closest to subcutaneous adipose tissue that was examined in our study (R=0.11, p=1.1*10^‑40^). Despite the clear significance of these correlations, the reported values of the correlation coefficients could be considered to be only modest. It should, however, be taken into account that two evolutionarily widely separated species and not exactly matching tissues were compared.”

[Editors' note: further revisions were suggested prior to acceptance, as described below.]

We are glad to see that you smoothed your HPA stress hypothesis according to our suggestions, but further revisions are still necessary to address our previous concerns about data quality and robustness of results have and remove the overinterpretation from the title.Specifically you need to:1) Control the batch effect using established packages, such as "BatchQC" or "BEclear";

As suggested, we analyzed potential batch effects systematically using BatchQC, added a respective source data set 2h with all reports for full transparency, summarized them in a paragraph in the *Control analyses* section and referenced the issue in the first paragraph of the *Results*.

“The number of samples collected and processed for this work required splitting them into different batches for sequencing. To analyze the degree this procedure potentially biased our results, we systematically investigated potential batch effects. Our results suggest that the sequencing strategy has little effect on the outcomes reported in the following (see Control analyses).”

“Control analyses: For practical reasons, samples collected for this project over several years were sequenced progressively in different batches. Therefore, we analyzed our data and sample scheme across all tissues and both species using BatchQC (Manimaran et al., 2016). In summary, the extensive search did not reveal any possible batch effects in the dataset from *F. micklemi*, although they may not be ruled out completely for *F. mechowii* (Source data 2h). A source for the minor batch effects in *F. mechowii* may be attributable to the repeated sequencing of the same sample(s) in different sequencing experiments – an approach avoided during generation of the *F. micklemi* dataset. However, because we followed a robust multifactorial analysis, in which differences must occur consistently across both species to be considered (see “Differentially expressed genes analysis” in Materials and methods), false-positive results due to batch effects are substantially reduced.”

2) Detect DEG using limma with voom to account for large variances and compare the results with DESeq2's output (see PMID: 23497356);

As suggested, we searched for DEGs using limma/voom. The respective p- and FDR-values obtained with this method are juxtaposed to the respective DESeq2 values in the tables in Source data 1e-g. We compare DESeq2 and limma/voom results across single tissues in the new Figure 2—figure supplement 2 and summarized this cross-validation in the following paragraph as part of the new *Control analyses* section.

“Control analyses: In addition to detecting DEGs with DESeq2 (Love et al., 2014), we also examined the central contrast, i.e., the juxtaposition of workers and breeders, using the combination of limma (Smyth, 2004) and voom (Law et al., 2014). The absolute numbers of DEGs found in each tissue, and thus the highly uneven distribution, were nearly identical for all tissues except thyroid (Figure 2—figure supplement 2). For the latter, however, a closer look shows that 85% of the DESeq2 DEGs (threshold FDR <0.05) are confirmed if the FDR threshold for limma/voom is raised from 0.05 to 0.1. If an FDR threshold of 0.05 is applied for both approaches across all tissues, a total of 58% of the DESeq2-DEGs are confirmed by limma/voom. Notably, all genes mentioned in the manuscript, including the candidates from Table 1, are among this set. Using a less strict threshold of 0.1 for limma/voom, 91% of DESeq2-DEGs are confirmed.”

Discussion: “For further validation, we applied the combination of limma (Smyth, 2004) and voom (Law et al., 2014) as an alternative DEG detection method to DESeq2 to account for potential dispersion problems in our data. All single DEGs mentioned in the manuscript were confirmed by this validation. This specifically includes all genes supporting our HPA hypothesis, e.g., *SULT2A1* (liver), *MC2R* (adrenal gland), *CYP11A1* (adrenal gland and ovary), *GH* (pituitary gland) and *IGF1* (ovary). In a similar way, we checked the main findings obtained with our standard bioinformatic p-value based enrichment approach (Fridley et al., 2010; Evangelou et al., 2012; Poole et al., 2016), using a second method that aggregates fold-changes instead. In particular, this step confirmed our initial finding of an enrichment of genes involved in the steroid hormone biosynthesis and ribosome metabolism as a central part of up-regulated anabolism in breeders, consistent with our HPA hypothesis (Figure 4—figure supplement 11) (see Control analyses).”

3) Redo the pathway enrichment with changes of expression values or other static indicators (see PMID: 23625889 and PMID: 32515539 for examples) but not p-value; to remove the bias and validate the HPA hypothesis.

The requested analysis was added to the *Control analyses* section and illustrated in a new supplemental figure. We refer to this additional check in our Results section. However, we would like to emphasize at the aggregation of p-values for the purpose of pathway enrichment in the context of gene expression studies has been used or even recommended in numerous previous studies (e.g. https://journals.plos.org/plosone/article?id=10.1371/journal.pone.0012693, https://www.tandfonline.com/doi/abs/10.1080/19466315.2014.888013, https://bmcbioinformatics.biomedcentral.com/articles/10.1186/1471-2105-14-368, https://bmcgenomics.biomedcentral.com/articles/10.1186/1471-2164-8-96, https://www.ncbi.nlm.nih.gov/pmc/articles/PMC3632109/).

Results: “As standard approach, we used an established method (Fridley et al., 2010; Evangelou et al., 2012; Poole et al., 2016) combining all p-values of genes in a given pathway in a threshold-free manner (Figure 4). […] In addition, we applied a second enrichment method that aggregates fold-changes instead (Figure 4—figure supplement 11; for a comparison of the results of both approaches, see *Control analyses* section).”

Discussion (see issue 2, above).

Control analyses: “In order to avoid potential biases by using solely p-values as indicator of differential gene expression, we additionally resorted to fold-changes as a test statistic for our pathway enrichment analyses. This exercise roughly confirms half of the pathways/hallmarks identified as differentially expressed at the cross-tissue level. Notably the set includes, consistent with the HPA hypothesis, steroid hormone biosynthesis and ribosomes as a central part of up-regulated anabolism in breeders (Figure 4—figure supplement 11). Furthermore, myogenesis, the P53 pathway, and coagulation were confirmed as cross-tissue differentially expressed by this approach. On the other hand, proteasome and oxidative phosphorylation gene sets, were not identified by the fold-change method.”

We have extended the corresponding Materials and methods section with the respective formulas. Also, other analyses pertaining to pathway enrichments were moved to the “Control analyses” section.

4) Reconsider the title, which is still an over-interpretation of the data and therefore needs to be fixed as follows. The words "slower aging" imply that the same biological aging process is occurring in the breeders and non-breeders, which is not actually clear from the data. Indeed, it seems quite plausible that the non-breeders die prematurely due to too much stress (as you speculate). Whether the rate of aging is different or not remains unclear. Replacing "slower aging" with "Increased longevity" or something like that would address this issue.

As suggested, we changed the title to “Increased longevity due to sexual activity in mole-rats is associated with transcriptional changes in the HPA stress axis”.